# Polysaccharides—Naturally Occurring Immune Modulators

**DOI:** 10.3390/polym15102373

**Published:** 2023-05-19

**Authors:** Emma J. Murphy, Gustavo Waltzer Fehrenbach, Ismin Zainol Abidin, Ciara Buckley, Therese Montgomery, Robert Pogue, Patrick Murray, Ian Major, Emanuele Rezoagli

**Affiliations:** 1Shannon Applied Biotechnology Centre, Midwest Campus, Technological University of the Shannon, V94EC5T Limerick, Ireland; emma.murphy@tus.ie (E.J.M.); patrick.murray@tus.ie (P.M.); 2LIFE–Health and Biosciences Research Institute, Midwest Campus, Technological University of the Shannon, V94EC5T Limerick, Ireland; 3PRISM, Research Institute, Midlands Campus, Technological University of the Shannon, N37 HD68 Athlone, Ireland; gfehrenbach@research.ait.ie (G.W.F.); i.izabidin@research.ait.ie (I.Z.A.); c.buckley@research.ait.ie (C.B.); imajor@ait.ie (I.M.); 4Applied Polymer Technologies, Midlands Campus, Technological University of the Shannon, N37 HD68 Athlone, Ireland; 5School of Science and Computing, Atlantic Technological University, H91 T8NW Galway, Ireland; therese.montgomery@atu.ie; 6Universidade Católica de Brasilia, QS 7 LOTE 1-Taguatinga, Brasília 71680-613, DF, Brazil; repogue@gmail.com; 7Department of Emergency and Intensive Care, Fondazione IRCCS San Gerardo dei Tintori, 20900 Monza, Italy; 8School of Medicine and Surgery, University of Milano-Bicocca, 20900 Monza, Italy

**Keywords:** polysaccharides, immune modulation, inflammation, naturally occurring polymers

## Abstract

The prevention of disease and infection requires immune systems that operate effectively. This is accomplished by the elimination of infections and abnormal cells. Immune or biological therapy treats disease by either stimulating or inhibiting the immune system, dependent upon the circumstances. In plants, animals, and microbes, polysaccharides are abundant biomacromolecules. Due to the intricacy of their structure, polysaccharides may interact with and impact the immune response; hence, they play a crucial role in the treatment of several human illnesses. There is an urgent need for the identification of natural biomolecules that may prevent infection and treat chronic disease. This article addresses some of the naturally occurring polysaccharides of known therapeutic potential that have already been identified. This article also discusses extraction methods and immunological modulatory capabilities.

## 1. Introduction

Biomaterials have substantially disparate functions and structural properties. They have played a critical role in a wide range of healthcare challenges, fostering significant advancements in several biomedical sectors, such as tissue engineering, medical implants, drug delivery, and immunotherapies [1,2,3]. Numerous healthcare products use biopolymers either as functional excipients or as active ingredients [4].

Polymers are a large group of natural or synthetic compounds that are comprised of macromolecules. These macromolecules in turn are composed of smaller monomers. Biopolymers are natural polymers that exist in nature, and include proteins, polysaccharides, cellulose, natural rubber, silk, and wool. Polymers can also be biobased, manufactured artificially from natural sources [5]. Synthetic polymers include nylon, polyethene, polyester, Teflon, and epoxy. Polymeric biomolecules are typically classed based on their monomeric unit [6,7,8,9].

Natural polymers are often favoured over synthetic ones for many reasons. They are abundant in nature, thus more accessible, and are easier to modify chemically. Additional advantages are that they are renewable, non-toxic, stable, hydrophilic, biocompatible, and biodegradable [10,11]. Synthetic polymers are often expensive with associated environmental and toxicity issues as well as lengthy synthetic processes. Classifications of natural polymers include polysaccharides, polypeptides, and polynucleotides [9]. This article will focus on one category: polysaccharides.

## 2. Polysaccharides

Polysaccharides are abundant biomacromolecules that are produced by plants, animals, and microorganisms. In addition to nucleic acids and proteins polysaccharides have a profound capacity for carrying biological information due to the complexity of their structure. Polysaccharides are the most naturally abundant biological macromolecules [12,13]. They are extended, high-molecular-weight polymers of monosaccharides held together by glycosidic linkages. Glycogen, cellulose, and starch are the most abundant polysaccharides known [14].

Structurally they are polymeric carbohydrates consisting of more than ten monomers connected by glycosidic bonds. Naturally occurring polysaccharides are abundant in structural diversity. Broadly, there are two groups of polysaccharides: homopolysaccharides which contain just one type of monomer, and heteropolysaccharides contain more than one type of monomer. Each source of polysaccharides has distinct branched chains, monosaccharide content, molecular weight (MW), and structural conformations [15].

Several polysaccharides are utilized for pharmaceutical purposes such as drug delivery. Other applications include encapsulation [16,17], gene therapy [18,19], wound healing applications [20,21,22], tissue engineering [23,24] and in the preparation of contact lenses for drug delivery [25].

Studies have demonstrated that polysaccharides in natural products offer a variety of health-promoting and therapeutic properties. This includes immunomodulatory, anti-cancer, anti-inflammatory, regenerative properties, and metabolic effects [26,27,28].

Polysaccharides can interact and influence the immune response and therefore play a vital role in the treatment of many human diseases [29]. There are variations in availability among sources. For example, starch, cellulose, and pectin are prevalent in plant tissue [30,31]. Yeast and Mushrooms are abundant in β-glucan polysaccharides [32,33]. Agar, carrageenan, and laminarin are some examples of marine polysaccharides. Lentinan, xanthan, dextran, curdlan, and pullulan are all examples of microbial polysaccharides. Polysaccharides are utilized by cells for structure, storage, adhesion, and cell recognition. Polysaccharides can also be secreted by micro-organisms, classified as exopolysaccharides. Their function encompasses mainly protection, adhesion and signalling [34,35,36].

There is a critical need to not only identify but also further characterize natural biomolecules that can prevent infection and alleviate chronic illness [37,38]. Due to their non-toxic nature and large effect on cellular functions, particularly immune cells, interest in naturally derived polysaccharide- polymers is increasing [15,39,40], Figure 1.

Thus, this review article will discuss some of the naturally available polysaccharides of known therapeutic potential, their extraction procedures, and potential immune-modulatory properties.

### Polysaccharide Classification

To date, over 300 distinct classes of naturally occurring polysaccharide molecules have been characterized. Polysaccharide structures are made up of condensed polymeric chains with monosaccharide units joined together by O-glycosidic linkages. They are characterized by their chemical structure, which consists of monosaccharide units connected by glycosidic linkages.

Monosaccharides, disaccharides, oligosaccharides, and polysaccharides are the classifications for carbohydrates which are based on molecular size or degree of polymerisation (DP). The chiral, polyhydroxylated aldoses or ketoses (monosaccharides) are incapable of being degraded into smaller carbohydrate molecules. Monosaccharides can be classified depending on the number of carbon atoms in their structure (pentose, five carbons), their carbonyl group composition (aldose) and their stereochemistry—spatial arrangements [41,42].

The composition of monosaccharides can be either linear or branched and can range in size. The DP, which is correlated to source, can range from 7000 to 15,000 for cellulose and up to more than 90,000 for amylopectin. Homopolysaccharides are homoglycans formed of the same monosaccharides, whereas heteropolysaccharides are heteroglycans composed of various monosaccharides. Starch, glycogen, and cellulose are examples of polysaccharides that are homopolysaccharides, while arabinoxylans, glucomannans, and hyaluronic acid are examples of heteropolysaccharides, which have two or more distinct types of sugar residues in their structure [14].

Based on their electrical charge, polysaccharides may be further classified as cationic (chitin, chitosan), anionic (heparin, hyaluronic acid, alginic acid, and chondroitin sulphate), non-ionic (chitin, chitosan, starch), amphoteric (carboxy-methyl-chitosan) and hydrophobic (cetyl-hydroxy-ethyl-cellulose) [43,44,45].

Polysaccharides are further differentiated by monosaccharide components, chain length, and branching. The glycosidic connection between the donor and acceptor forms can be linear or branched chain, unlike proteins and peptides, which only have linear chains. Polysaccharides store energy (eg. starch) in cells or offer structure and stability (eg. cellulose) [46].

Polysaccharides that are found in terrestrial plants consist of three networks that are structurally distinct from one another. Firstly, there are microfibrils of cellulose that have been coated with adhering non-cellulosic polymers such glucuronoarabinoxylans, glucomannans, and xyloglucans. Secondly, the gel-like matrix, which is composed of pectin and other polysaccharides. Thirdly, there are structural proteins that are cross-linked to polysaccharides [47,48].

Algal polysaccharides illustrate the enormous structural variations between polysaccharides from similar sources. Algal polysaccharides are non-toxic, inexpensive, biodegradable, and biocompatible natural polymers [49]. Polysaccharides from this source contain sulphated esters and are thus classified as sulphated polysaccharides [50]. Sulphated polysaccharides can be identified by the presence of sulphate groups. Sulfation of polysaccharides causes changes in steric hindrance and electrostatic repulsion in the sulphated hydroxyl groups. These changes result in bending and extension of the chain as well as an increase in water solubility. Such dynamics, in the end, are what contribute to their power to change biological activity [51]. Sulphated polysaccharides possess immunomodulatory activity and have been identified in several microalgae and macroalgae plants of both marine and freshwater [52,53,54].

Marine algae can be classified into three groups according to thallus color. The three categories are: brown, red and green. Thallus color is derived from natural chlorophylls and pigments.

Polysaccharides found in brown seaweeds (Phaeophyta) include laminarin, alginate, and fucoidan. This polysaccharide contains glucose, rhamnose, galactose, fructose, and xylose monomers. [55,56,57]. Red seaweeds (Rhodophyta) include polysaccharides such as carrageenan agar and agarose. Monosaccharide composition contains glucose, galactose, and 3,6-anhydro-galactose [52,53,54]. Green algae (Chlorophyta) include cellulose, mannan, sulphated rhamnan, and ulvan polysaccharides. Monosaccharide subunits include glucose, mannose, rhamnose, xylose, iduronic acid, and glucuronic acid [58,59,60]. With structural variation comes a variety of extraction processes aimed at isolating the component of interest while preserving its basic form.

## 3. Extraction of Polysaccharides

The choice of extraction techniques is determined by the physicochemical qualities of the components, the extraction environment, and the presence of interfering compounds. The extraction conditions should be based on the structural components of the polysaccharide types, which are mostly discovered in the cell walls of diverse plants, fungi, and algae. The objective of the separation and purification of polysaccharides is to preserve their natural qualities [61] and, more importantly, their biological activity. Extraction techniques are outlined in Table 1.

### 3.1. Extraction of β-Glucans

Water, physical methods (ultrasound and radiation), chemical techniques (hot alkali and acid–alkali), natural methods (enzyme process), or a combination of these approaches may be used to extract crude β-glucan. As β-glucan is often a structural component, it is often crosslinked to other molecules. Enzymes and polysaccharide purification using column chromatography or ultrafiltration are all used in the deproteinization process [80,81,82,83,84], Figure 2.

To fully understand the correlation between β-glucans and activity, assays should be carried out to measure bioactivity after extraction. Although expensive, ideally, chemical characterization should also be carried out using techniques such as NMR.

Hromadkova et al. (2003) extracted particulate β-glucan from *S. cerevisiae* cell walls by solvent exchange, lyophilization, and spray-drying. Mitogenic and comedogenic tests assessed β-glucan samples’ immunological activity. Differentially dried particle β-glucan samples demonstrated twice the immunomodulatory activity of lyophilization and solvent-exchange-derived samples. When utilizing particle 1,3-β-glucan as an immunomodulator/adjuvant in an aqueous solution, the authors propose using spray-dried formulations [85].

Li et al. (2019) developed an extraction technique to increase the amount of β-glucan that could be extracted from *Lentinus edodes.* To efficiently extract the pure polysaccharides, an alkaline solution was used. The extraction method separated components into α- and β-configurations separately. The method demonstrated that 0.1 mol/L to 0.5 mol/L NaOH alkaline solution was sufficient to extract β-glucan [62].

In this process, *L. edodes* fruiting body powder was extracted using water (20:1 *v*/*w*) at 25 °C for 5 h, and the residue was dried. The alkaline extraction was performed using a sodium hydroxide (NaOH) solution at a 20:1 (*v*/*w*) ratio. Parameters tested included extraction temperatures of 0, 20, and 60 °C; extraction timepoints of 0.5, 1, and 2 h; and sodium hydroxide concentrations of 0.1, 0.25, and 0.5 mol/L. For extraction, the alkaline-soluble crude polysaccharides were extracted from the water-extractable residue. The supernatant was then neutralized, dialyzed, and precipitated (1:4 *v*/*v*) in ethanol. Subsequently, the extract was freeze-dried. The extract was then resuspended in 15 mL of deionized water, and the resulting solution was loaded onto a Q-Sepharose Fast Flow strong anion-exchange column (QFF, 4.6 15 cm). At a flow rate of 4 mL/min, the polysaccharide was eluted with 600 mL of distilled water and 0.2 and 0.4 mol/L sodium chloride (NaCl) solutions [62]. The methylation and NMR analysis showed that the polysaccharide extract was structurally described as a 1,6 β-glucan with trace levels of 1,3-d-glucosidic side chains [62].

According to Amer et al. (2021), a modified acid–base extraction method will yield β-glucan extracts from *Saccharomyces cerevisiae* with a higher biological efficacy compared to extracts from a water-based extraction method. This was demonstrated by antimicrobial activity against multidrug-resistant bacteria, fungi, and yeast [63]. 

Immunomodulatory effects were observed through strong induction of TNFα, IL-6, IFN-γ, and IL-2. Yeast cells were grown for 48 h in a yeast-extract–glucose-broth medium. The yeast cell pellets were collected, mixed with five-fold 1 M NaOH, heated for two hours at 80 °C in a water bath, and centrifuged again in distilled water. Samples were then centrifuged to obtain the final yield. The pellets of lysed yeast cells were then collected for β-glucan extraction. The extraction was carried out five-fold, using acetic acid (CH_3_COOH). The collected pellets of β-glucan were dried in an oven at 60 °C to eliminate any residues of organic-soluble components, excess proteins, or other possible contaminants. The extracts were characterized using FTIR and HPLC [63]. 

β-glucans can be obtained from an abundance of sources. Each extraction method is specific to the tissue being lysed or hydrolyzed for extraction. As there is no defined universal method for extraction, extracts should be characterized to ensure that the correct compound is isolated.

### 3.2. Extraction of Fucoidan

As with most of the polysaccharides discussed here, fucoidan extraction conditions influence biological activity [86,87]. Low molecular-weight fucoidan was found to have higher anticancer activity in comparison to higher-weight fucoidan. Enzymatic degradation was also found to increase anticancer properties [88]. Figure 3.

Fucoidans must be extracted with as few interfering molecules as possible. Fucoidan extraction strategies including acidic, hot water, and organic solvents, depending on cell wall polysaccharide solubility. These procedures can be time-consuming, toxic, inefficient, and may impair fucoidan’s fine structure and bioactivity [89]. More desirable, novel, green extraction techniques include supercritical water extraction [90], microwave-assisted extraction [91] and ultrasound-assisted extraction [92].

Enzymatic approaches are preferred as it is important to remove other molecules such as alginate that are crosslinked to cell wall structures [89]. However, enzyme-assisted extraction techniques for sulfated fucoidan polysaccharides required carbohydrase combinations that hydrolyze starch not present in brown seaweeds [93,94,95].

A study by Nguyen et al. (2020) potentially addressed this. Fucoidans were extracted from F. evanescens and S. latissima, using Cellic^®^Ctec2 from Novozymes and alginate lyase SALy from Sphingomonas sp. This enzyme-assisted extraction approach incorporates a one-step process at pH 6.0, 40 °C, elimination of non-fucoidan polysaccharides by Ca^2+^ precipitation, and ethanol-precipitation of crude fucoidan. Enzymatic fucoidan yields were equivalent to chemically extracted yields for F. evanescens and S. latissima, although molecular sizes were substantially greater [89].

To fully understand the complex mechanisms behind the structure–activity relationship of polysaccharides and immune cells, studies should measure activity after extraction. Furthermore, all polysaccharides should be characterized after extraction. Numerous studies have combined these important measurements.

Fucoidan from Saccharina japonica was extracted using subcritical water extraction (SWE) in a study from Saravana et al. (2017). Box–Behnken design (BBD) was used to optimize temperature, pressure, solid-to-liquid (S/L) ratio, agitation speed, and reaction time (RT). The best conditions were 127.01 °C, 80 bar, 0.04 g mL1, 300 rpm, and 11.98 min. The crude fucoidan (CF) yield was 13.56 percent. FTIR and UV–Vis identified the polysaccharides that were fucoidan. The extract demonstrated strong antioxidant and anti-proliferative effects. The study demonstrates that SWE may be the preferred approach for industrial-scale fucoidan synthesis due to its high yield and functional activity [90].

Microwave digestion is another green technology. Microwave-assisted extraction (MAE) was used to extract fucoidan from Ascophyllum nodosum. Different extraction parameters were investigated. Optimal fucoidan yield of 16.08% was achieved at 120 °C for 15 min. The sulphated fucoidan was characterized by GPC, HPEAC, and IR analysis. Fucose was the predominant monosaccharide extracted at 90 °C, while glucuronic acid was the main monosaccharide extracted at 150 °C. With decreased extraction temperature, the fucoidan molecular weight and sulphate content increased. All fucoidans had antioxidant activity, but 90 °C extracted fucoidan had the greatest. This work demonstrated that MAE is an effective way to extract sulfated polysaccharides from seaweed such as Ascophyllum nodosum, while still maintaining antioxidant activity [91].

### 3.3. Extraction of Glucomannan

In a study carried out by Wang et al. (2020), a water-soluble polysaccharide was extracted and purified from the pseudobulb Bletilla striata. In this method, the pretreated powder was extracted with water at 80 °C for 4 h. Then, 4 L of 95% ethanol was added to the solution at 4 °C for 24 h. The precipitate was redissolved in distilled water and then frozen and thawed 10 times to eliminate protein. The deproteinated material was dialyzed against distilled water for 72 h to remove small compounds, Figure 4.

The crude polysaccharides were re-dissolved in distilled water (10 mg/mL), put on a DEAE-52 cellulose column, and eluted in a stepwise gradient of 0 to 0.4 M NaCl at 1 mL/min. The phenol–sulfuric acid technique determined the sugar content. The eluent was then dialyzed, lyophilized, and purified using a Sephadex G-150 column (5.0 60 cm). It was determined that the extract was composed of only mannose (m) and glucose (n) at a molar ratio of 7.45:2.5. The molecular weight was found to be approximately 1.7 × 10^5^ Da and 1.11 × 10^5^ Da, respectively [96].

In terms of activity, the extract demonstrated gastroprotective efficacy in vivo, as well as protective action against ethanol-induced GES-1 cell damage in vitro. Particularly, a high dosage of 100 mg/kg may decrease the stomach mucosa’s ulcer index and raise the percentage of ulcer inhibition, which may have been produced by boosting the tissue’s antioxidant capacity and obstructing the apoptotic pathway. The extract also had comparable gastroprotective effects to the positive control (Omeprazole) [96].

C. barometz is a species of tree fern, an industrial crop in China. The aim of a study by Huang et al. (2018) was to determine the effects of a polysaccharide isolated from C. barometz (CBB) on rats that had undergone ovariectomy (OVX). For extraction, the C. barometz rhizome was cut into bits, steeped overnight in 1:10 deionized water, and extracted at 80 °C for 3 h. The residues were extracted with NaOH at room temperature for 3 h and then pooled. The extracts were filtered, neutralized with 0.3 mol/L hydrochloric acid, concentrated, and centrifuged.

Next, ethanol was added to the supernatants and incubated for 24 h at room temperature to yield crude saccharides, which were deproteinized using Sevag reagent. In the final step, the supernatants were dialyzed and lyophilized. The final extract was isolated and purified by DEAE-Cellulose 52 and G-75 gel filtration chromatography. The final extract was composed of glucose and mannose in d-configuration. When the extract was administered to ovariectomized rats, osteoprotective effects were observed: both bone mineral content (BMC) and bone mineral density (BMD) were increased [97].

In another study, hydrochloric acid was used to hydrolyze starch in porang flour to obtain glucomannan. Crude porang flour was dispersed in hydrochloric acid solution at room temperature. The solution was heated in a water bath heater while being stirred. Continuous stirring allowed acid hydrolysis of starch and glucomannan extraction at a suitable temperature. Diluting the residual reaction mixture with demineralized water and centrifuging (9000 g, 30 min, and 298 K) removed insoluble materials. After rotary evaporation, the filtrate volume was reduced by a third. Glucomannan was precipitated overnight by adding 95% ethanol at 277 K and centrifuging (9000 g, 40 min, and 298 K). The pellets were washed twice with anhydrous ethanol, vacuum-filtered, and then freeze-dried for 48 h. Milling and sieving the dry material yielded refined glucomannan [98].

### 3.4. Extraction of Chitosan

The preparation of chitosan mainly includes three stages: (i) demineralization, (ii) deproteinization, and (iii) deacetylation. Initially, the waste material is demineralized to remove inorganic matter by applying dilute HCl (up to 10%) with agitation [99,100]. The second stage of deproteinization removes the protein and other organic components in the shells by reacting with heated alkali solution, such as 1–10% (*w*/*w*) NaOH or KOH [99]. Currently, there is a preference for the extraction and purification of chitin by lactic acid fermentation for deproteinization and demineralization [100]. Figure 5.

In general, the process of chitin conversion into chitosan can be categorized into two types, either via chemical or biological (enzymatic) methods [99]. Deacetylation is the process of converting chitin to chitosan by removing acetyl groups from *N*-acetylglucosamine to form d-glucosamine units, which contain free amino groups and increase the solubility of the polymer in aqueous media. The chemical hydrolysis of the acetamide groups is achieved in a strongly alkaline medium and at high temperatures. Generally, the reaction is carried out in a heterogenous phase, using concentrated solutions of NaOH or KOH (40–50%) at temperatures above 100 °C, preferably under an inert atmosphere to avoid depolymerization of the polymer [99].

The reaction-specific conditions depend on factors such as the source of material used and the desired degree of deacetylation. Currently, enzymatic methods using chitinases or chitin deacetylases are a viable alternative to produce biologically active chitosan [100].

### 3.5. Extraction of Alginate

Alginates can be extracted in many ways depending on the desired application. The most used method relies on extracting the alginate as sodium alginate [101]. The process starts with a precipitation step by mixing the brown algae with calcium chloride or mineral acid. This step converts the insoluble alginate present in the cell walls, as calcium and magnesium alginate into insoluble calcium alginate fiber or alginic acid gel, respectively [102,103], Figure 6.

This is then followed by a neutralization step by mixing either sodium carbonate or sodium hydroxide to produce water-soluble sodium alginate [102]. Other types of alginate salt, such as calcium alginate, potassium alginate, and ammonium alginate, can be produced by adding an appropriate alkali solution, usually calcium chloride, potassium carbonate, or ammonium hydroxide, to the acid gel [102].

Although this is the most common method, it poses some disadvantages. As the alginate dissolves as sodium alginate, the thickness of the solution hinders filtration, and the solution must be diluted with large quantities of water. Filter aids required for this method make the process expensive, and the dilution might influence the physicochemical properties of alginates [101].

As an alternative, enzyme-assisted extraction techniques are employed. This method involves degrading the cell walls by using enzymes to extract the desired compounds. Many studies have reported using β-d-mannuronate and α-l-guluronate lyases that eliminate the (1→4) glycosidic bond connecting the two uronic acid units, thus degrading the chain [101]. Some studies have also used different types of carbohydrase and proteases to catalyze the degradation of the cell wall [104,105].

### 3.6. Extraction of Hyaluronic acid

While some bacterial species naturally produce HA, there are significant caveats, such as the presence of endotoxins. Therefore, the preferred approach is the biotechnological production of HA. Strains such as *Lactococcus lactis*, which are GRAS strains, have been bioengineered using the HA synthesis operon and the lacF selectable marker to produce HA via fermentation. Because of the lack of any hyaluronidase activity in *L. Lactis* [106] it is a particularly good candidate for HA production, Figure 5.

After induction of HA production, capsular HA can be extracted through the addition of detergent to the medium followed by centrifugation. The yields obtained in strains such as *L. lactis* tend to be lower than that achieved in *Streptococcus* spp. [107], however, researchers have been achieving higher yields through the addition of nisin-induced expression systems.

HA extracted from animal tissues also remains significant in industrial production due to the high molecular weights recovered, as detailed in Table 1. One of the main sources of animal-derived HA is rooster comb, with variable yields being reported in the literature [108]. The variations in extraction method leads to great variation in yield and polydispersity of molecular weights obtained due to the harsh mechanical and solvent processes utilized. Typically, frozen rooster comb would be ground in an electric grinder to yield pieces of approximately 0.5 cm in size. To remove the fat from the rooster comb, the pieces are placed in a vat of acetone in a refrigerator, with regular solvent replacements. Once the fat has been sufficiently removed, the acetone is evaporated, and the dried rooster combs are extracted in 5% sodium acetate solution. The aqueous extract obtained is then diluted in ethanol, and the precipitate is centrifuged and dissolved in 5% sodium acetate and centrifuged again.

Impurities such as contaminating proteins, which could illicit an immune response, are removed by shaking with chloroform several times followed by chloroform-amyl alcohol several times until a gel no longer forms. Finally, the solution is dialyzed and precipitated with ethanol before lyophilization [109]. Because of this lengthy extraction process, which yields variable results and has high associated costs and labor, biotechnological solutions are the preferred route of extraction.

### 3.7. Extraction of Carrageenan

CRG is obtained from various genera in the *Florideophycea* class, such as *Chondrus, Furcellaria, Eucheuma, Agardhiella, Gigartina, Iridaea, Sarconema,* and *Hypnea.* They can be harvested from the North American, French, and Iberian Peninsula coasts or cultured in shallow waters, as is performed in The Philippines, Indonesia, Madagascar, and the east coast of Africa [110], Figure 7.

CRG oligosaccharides are obtained via hydrolysis, alkali extraction, and boiling methods and, based on the purity level are classified as refined CRG (RCRG) and semi-refined CRG (SRCRG). RCRG is extracted by alkali solutions, boiling water, or enzymatic reactions, while SRCRG, the CRG, is not extracted from the seaweed, using simpler methods such as hot alkali digestion [111].

In general, most CRG oligosaccharides can be fractioned according to sulphation level (κ, ι, and λ) by enzymatic, chemical, and physical extraction to obtain oligosaccharides with reported biological activities such as antioxidant, antitumor, anti-inflammatory, and immunomodulatory activities [112].

Enzymatic methods have higher yields than traditional boiling methods. The reaction is based on the digestion of the seaweed cell wall, where cellulose is the main structural polysaccharide, and it also contains CRG. Cellulose is lysed by cellulase and releases CRG, without affecting its structure [111].

Chemical methods use reactive chemicals such as H_2_O_2_ and diluted acids such as HCl or H_2_SO_4_, with a lower extraction cost when compared to enzymatic extraction and a higher environmental impact compared to other alternatives. Physical methods are based on microwaves and irradiation disruption, using ultra-sound, microwave, ultraviolet, and gamma irradiation [112]. The extraction yield of physical methods is lower when compared to enzymatic and chemical processes, but it can be used to support those methods by increasing yields, reducing the volume of required reagent, and consequently minimizing the environmental impact.

### 3.8. Extraction of Ulvan

Ulvan’s composition is highly dependent on culture and abiotic factors that must be considered when selecting the extraction/purification techniques. In general, these polysaccharides are extracted in hot water at 80–90 °C and even at 120 °C since no degradation of sulfation or polymerization is observed. However, temperatures as high as 120 °C increase the interactions between cell wall components and ulvan, leading to lower yields. The use of acids and chelators such as EDTA to remove divalent cations is an alternative to overcome this problem and enhance the disruption of the cell wall. Then, ulvan polysaccharides are precipitated with ethanol or the addition of a quaternary ammonium salt [60].

Hydrochloric acid (HCl) and sodium hydroxide (NaOH) can be used to obtain acid and alkali polysaccharides, respectively, or enzymatic digestion and chromatography can be employed to obtain highly specific products [113]. The duration of extraction and pH are important factors to be considered. Based on the data available in the literature, Kidgell et al. (2019) recommended extraction at 80–90 °C, a pH of 2–4.5, and a duration of 1–3 h for a high extraction yield, high selectivity, and the low degradation of ulvan polysaccharides [114], Figure 6.

### 3.9. Extraction of Xylans

The isolation of xylans from hardwoods and non-endospermic tissues of plants includes various multistep extraction procedures. In the case of hardwoods, delignification with acidic sodium chlorite is usually used before the alkaline extraction of xylan [114]. There have been studies that have successfully extracted xylan with processes that involve an alkaline and H_2_O_2_/NaOH delignification step and a subsequent alkaline extraction step [114], Figure 2.

There have been reports that the treatment of the crude materials in various reagents prior to the alkaline procedure, such as aqueous ethanol and concentrated ammonia, can significantly increase the extractability of xylan [115].

Alternatively, some studies used steaming or hot water treatment methods on the plant source instead of the alkaline procedure. These methods relate to the degradation and solubilization of hemicelluloses, yielding oligomeric products and low-molecular-weight xylans [114].

Compared to wood, the extractability of xylan from lignified plant fibers is easier potentially due to the lower amounts and different structure of the lignin component, as well as different cell and plant architectures. Milder extraction conditions and even oxidant-free dilute alkalis can be used to yield more than 85% of the xylan components of corncobs and wheat [114].

A study was conducted comparing the extraction of xylan from wheat bran by alkaline and enzymatic methods. It was reported that a 50% greater yield was obtained by using the alkaline method. However, the immune-enhancing activity of extract obtained from the enzymatic method was higher [116]. Therefore, it can be summarized that, to choose an extraction method, it is important to decide which xylan type will be produced and the properties that are suitable for which applications it is needed.

## 4. Immune Modulation

Immune systems that function effectively are essential for the prevention of disease, elimination of infection, and maintenance of homeostasis of the organism. This is achieved through the effective removal of pathogens and irregular cells or debris.

The human immune system has two branches: the innate immune system, which produces a non-specific inflammatory response in response to the instant identification of foreign material; and the adaptive immune system, which conducts highly specific antigen responses and maintains a long-term memory [32].

The innate system consists of polymorphonuclear cells, mononuclear phagocyte cells (dendritic cells—DCs; monocytes; and macrophages), and lymphocytes (natural killer cells, gamma delta T-cells, and innate lymphoid cells). The adaptive system consists of B and T lymphocytes. Using soluble substances and cellular subsets, the establishment of an adequate and efficient immune response needs tight, coordinated, and carefully managed crosstalk between the two systems [117]. Bioactive molecules can facilitate crosstalk through immune modulation.

Although necessary for life, the immune response can malfunction. Immune system dysfunction can be classified into four broad categories. Primary immune deficiency occurs when an individual is born with a weakened immune system. When a disease impairs the immune system, this is referred to as acquired immune deficiency (e.g., HIV-mediated immunodeficiency). Autoimmune illness occurs when the immune system erroneously targets healthy tissues (e.g., rheumatoid arthritis). Finally, an overactive immune system can lead to autoimmunity, allergies, and inflammatory illnesses, whereas an underactive immune system can lead to infections and malignancies [118].

For optimal health, the immune system must be strictly regulated. Autoimmune disorders and associated inflammatory diseases can occur because of hyperactivity, or when immunity is triggered against self-antigens or harmless substances. In contrast, chronic infections and malignancies can emerge when the immune system is poorly engaged during inflammation [118].

Innate immune cells perceive pathogen- and damage-associated molecular patterns through germline-encoded pattern-recognition receptors (PRRs). Rapid, non-specific actions initiated by PRRs include phagocytosis, cell motility, pathogen or cell death, and cytokine synthesis. These innate immune responses eliminate invading microorganisms effectively [119].

PRR signaling pathways have been extensively defined as initiators of cascades that eventually lead to leukocyte migration to the site of infection. The recruitment of leukocytes is necessary for inflammation [120]. The defensive response of living tissues with a circulatory system to infectious/predatory stimuli and local damage is known as inflammation.

When the immune system is activated, the body releases inflammatory cells. This process helps to eliminate unwanted stimuli or pathogens and promote healing, in contrast to the acquired immune system, which targets in a more specific fashion. Inflammation is also a symptom of several disorders. The immune system must maintain an optimal redox balance; failure to do so may result in the pathogenesis of immune-related disorders inflammatory bowel disease, dermatitis, metabolic syndrome, asthma, and acute respiratory distress syndrome [121,122,123]. These reactions can be modulated by bioactive substances that activate or suppress immune cell function.

Polysaccharides have several functional groups, diverse physicochemical characteristics, and significant biological activities, making them excellent materials for a range of therapeutic applications [124,125]. Polysaccharides have been identified to modulate a dysregulated immune system [126,127,128].

Polysaccharides generated from natural sources have many regulatory effects on the immune system. They do this by activating complement and boosting cytokine production [129,130,131]. Research suggests that polysaccharides, namely those from microbial sources, are recognized as pathogen-associated molecular patterns (PAMPs). These patterns are recognized through PRRs and elicit their effects through this mechanism. The exact pathways and structure–activity relationship are still not fully understood [33]. Some of the immune-modulating properties of polysaccharides are outlined in Table 2.

There are several commercially available polysaccharides from various sources for the treatment of various diseases. Poria PS is an oral solution which has anti-tumor, anti-inflammatory, anti-hepatic and anti-diabetic properties [132]. Microbial PS Vi polysaccharides are used in pneumococcal vaccines [133]. Xanthan gum, gellan gum, and scleroglucan polysaccharides isolated from microbial sources have been used for drug delivery systems, [134] as drug absorption can be increased and recognized by immune counterparts when combined with polysaccharides. Other polysaccharides such as chitosan can have low immunogenic properties and therefore are more suitable for wound healing and bone regeneration applications [135,136].

The functional activity of polysaccharides is strongly correlated to structure. Stability, hydrophilicity, and biodegradability, coupled with other physicochemical features of natural polysaccharides, support their wide variety of biological activities [137,138]. According to Kralovec et al., a polysaccharide/glycoprotein complex mostly composed of galactose, rhamnose, and arabinose with a molecular weight >100 kDa displays potent biological activity. Researchers have also hypothesized that the four most important monosaccharide components contributing to macrophage stimulating activity are arabinose, mannose, xylose, and galactose, while the most common monosaccharide component, glucose, played no clear role in the immunoregulatory activity of polysaccharides [139,140,141].

**Table 2 polymers-15-02373-t002:** Immunomodulatory activity of polysaccharides from natural sources.

Polysaccharide	Model	Immuno-Modulatory Effect	Reference
**Fucoidan**	In vitro cell culture,	Activates NF-κB in macrophages (in vitro). Promotes maturation of dendritic cells. Inhibits the polarization of macrophages toward the tumor-promoting M2 phenotype (in vitro).	[142]
**Fucoidan *Undariia pinnatifilda***	In vitro cell culture—neutrophils.	Delayed spontaneous apoptosis is associated with increased levels of anti-apoptotic protein Mcl-1 and decreased levels of activated caspase -3. Induced activation of neutrophil–secretion of IL-6, IL-8, and TNF-α–AKT-dependent manner.	[143]
**Fucoidan *Macrocystis pyrifera***	In vitro cell culture—human neutrophils, mouse NK cells, spleen dendritic cells, and T-cell in vivo mouse model.	Delayed human neutrophil apoptosis. Promoted NK cells activation and cytotoxic activity. Promoted DC maturation. Enhanced T-cell immune response, antigen-specific antibody production, and memory T-cell generation.	[115]
**Fucoidan** ** *Fucus vesiculosus* **	In vivo mouse model.	Upregulation of CD40, CD80, and CD86 expression. Production of IL-6, IL-12, and TNF-α in spleen DCs. Influenced INF-γ-producing Th1 and Tc1 cells in an IL-12-dependent manner.	[144]
**Fucoidan** ** *Clad siphon navae-caledoniae* **	In vitro cell culture. Estrogen-positive and estrogen-negative breast cells.	Co-treatments with chemotherapy drugs inhibited cell growth, apoptosis, and cell-cycle modifications. Decrease in phosphorylation of ERK and Akt. Enhanced intracellular ROS levels. Reduced glutathione levels.	[145]
**Fucoidan commercial powder 10 mg/mL**	Prospective open-label, single-arm clinical study.	Reduction in pro-inflammatory cytokines IL-1β.	[146]
**Fucoidan *Kjellmaniella crassifolia* and *Undaria pinnatifida***	In vitro macrophages RAW264.7. In vivo mouse model.	Enhanced cell proliferation enhanced the secretion of granulocyte-macrophage-colony-stimulating factor (GM-CSF) and tumor necrosis factor-α (TNF-α). Increased the secretion of GM-CSF, TNF-α, interleukin (IL)-4 and IL-10 in vivo.	[147]
* **Fucoidan Sargassum** * **species and** * **Fucus vesiculosus** *	Lewis lung carcinoma cells and melanoma B16 cells. In vivo mouse model.	Decreased the viable number of cancerous cells in a dose–response manner (in vitro). Cytolytic activity of natural killer (NK) cells was enhanced (in vivo).	[148]
**Chitosan**	Isolated spleens from oligodendrocyte glycoprotein (MOG) induced experimental autoimmune encephalomyelitis (EAE) mice.	Increases INF-γ and IL-10 levels.	[149]
**Chitosan**	Murine model of autoimmune encephalomyelitis.	Improvement in clinical signs. Reduction in demyelination. INF-γ, IL-17, and TNF-α levels reduced.	[150]
**Chitosan**	In vitro cell culture.	Induces production of TNF-α, IL-6, and INF-γ in macrophages. Promote the expression of the genes, including iNOS and TNF-α.	[151]
**Chitosan**	Leishmania infection in BALB/c mice.	Increases INF-γ secretion. Reduction in lesion formation. Lower parasite load.	[152]
**Alginate (Commercial)**	In vitro cell culture.	Increases TLR-4 expression. Activates NF-ĸB and MAPK pathways. Promote TLR-4-mediated phagocytosis.	[153]
**Xylans** **Corn cobs**	Dextran sodium sulphate (DSS)-induced UC mouse model.	Reversed the imbalance between pro- and anti-inflammatory cytokines. Rebalanced gut microbiota and reduced *Oscillibacter, Ruminococcaceae, Erysipelatoclostridium,* and *Defluviitaleaceae* nuclear factor-κB (NF-κB) inhibition. Reduction of inflammatory intestinal damage.	[154]
**Commercial λ-Carrageenan**	Melanoma B16-F10 and mammary cancer 4T1 mouse models.	Inhibited tumor growth. Enhanced tumor immune response. Increased the number of tumor-infiltrating M1 macrophages and dendritic cells. Enhanced the secretion of IL17A in spleen. Increased the level of TNF-α in tumor.	[155]
**Ulvan**	*Labeo rohita*	Increase in red blood cells and white blood cells. Increase in superoxide dismutase activity. Increase in respiratory burst activity. Increase in phagocytic activity. Increase in expression IL-1β, TNFα, hepcidin, and IgM cytokines.	[156]
**Hyaluronic acid**	Clinical trial Patients of knee osteoarthritis.	Lymphocyte number increased. Reduction of IL-6 and IL-8.	[157]
	In vitro cell culture–murine macrophages.	Nitric oxide production by LPS-stimulated macrophages was decreased. Decreased expression of TNF-α, IL-6, CCL2, and IL-1β in LPS-stimulated macrophages. Increased expression of TGF-β1, IL10, IL-11, and Arg1 genes/anti-inflammatory responses.	[158]
**Glucomannan** **Amorphophallus konjac**	Cyclophosphamide -induced immunosuppressive mice.	Enhanced the spleen indices. Enhanced the thymus indices. Reduced the proliferation of splenic lymphocytes. Enhanced and regulated humoral immune activity via serum hemolysin. Reduce the capacity of NK cell lethality. Reduction of phagocytic activity of peritoneal macrophages was extremely significantly reduced. Reduction in the production of IL-2, IgG, and TNF-α.	[159]
**Glucomannan** ** *Dendrobium officinale* **	In vitro cell culture.	Stimulate cytokine production (TNF-α and IL-1β). Induced immune activities involving ERK1/2 and NF-κB pathways.	[160]
**Glucomannan** ** *Dendrobium officinale* **	In vitro cell culture.	Promoted the degradation of IκB complexes and activated NF-κB phosphorylation.	[161]
**β-glucans Paramylon** ** *E. gracillis* **	In vivo mouse model (wound healing).	Increased wound contraction. Increased IFN-γ levels.	[162]
**β-glucans** **Paramylon** ** *E. gracillis* **	In vivo mouse model (influenza infected).	Increased survival rate. Lower virus titer compared to control group. Increased levels of cytokines-IL-1β, IL-6, IL-12, IL-10, IFN-γ, and TNF-α.	[163]
**β-glucans** **Lentinan** ** *E. gracillis* **	In vitro cell culture.	Increased levels of IL-6, TNF-α IL-22, IL-8, and IL-10 in THP-1 macrophages. IL-8 and TNF-α reduction after cytokine insult in A549 and BEAS-2B lung cells.	[128]

## 5. Natural Polysaccharides with Immune-Modulatory Activity

### 5.1. β-Glucans

β-glucans/Beta-glucans are a broad family of complex polysaccharides that are present in an abundance of sources. They are physiologically active molecules extensively shown to improve health [164]. Their origin determines whether they are categorized as cereal or non-cereal derived. Oat and barley are cereal sources of β-glucans, whereas mushrooms, algae, bacteria, and seaweed are examples of non-cereal sources [33]. Specific to this group of polysaccharides is a 1,3 beta-glycosidic-linked backbone. Separate from this, the polysaccharide can take many forms, dictated by origin, which ultimately influences activity. All β-glucans are homo-polysaccharides [165]. The structural variation is introduced at the branches that emerge from this central backbone. Both unbranched and branched β-glucans exist. The 1,4 or 1,6 locations are the branching points [166].

Classification is largely based on structure, since β-glucans derived from cereal have a distinct structure from those derived from non-cereal sources. Within this classification, β-glucans derived from cereal sources may also vary structurally; β-glucans derived from barley can be structurally distinct from oat β-glucans. The same variability exists among β-glucans derived from non-cereal sources. [167,168].

β-glucan polysaccharide structure is strongly linked to activity; variation will affect activity. The β-glucan lentinan, isolated from mushrooms has a β-helix conformation. This polysaccharide has anti-tumor properties [169,170]. When this helix is damaged, biological activity also decreases. Studies have also shown that activity is also correlated to molecular weight and chain conformation, which can be altered during extraction [171]. It is therefore important that the chosen extraction method does not alter the molecular structure.

#### Immuno-Modulatory Activity of Beta Glucans

Like extraction techniques, there are several immune-modulatory effects identified for β-glucans. Functional activity is determined by molecular and structural properties since β-glucans have a well-defined structure–activity relationship [172].

β-glucans obtained from cereals have metabolic effects. Modification of the gut flora, the lowering of cholesterol, and decreased cardiovascular and diabetes risk are among the reported activities. Immunomodulatory actions, antitumor effects, wound healing, and amelioration of immune-related disorders are all effects associated with non-cereal β-glucans. For example, during training, yeast β-glucan may improve athletes’ resistance to infection. It has been shown that adding baker’s yeast to a diet may delay the development of upper respiratory tract infections [173].

Consuming indigestible carbohydrates such as β-glucans may enhance the immune system, as they enter the large intestine largely intact. This is beneficial, as it is in this location that they encounter the immune cell type macrophages, an important cell in first-line innate defense. Macrophages exposed to β-glucans show enhanced phagocytic activity and an anti-inflammatory profile [174,175].

This property was well illustrated in a study by Tanioka et al. (2013), who demonstrated the effects of β-glucans from *Aureobasidium pullulans* on normal and immunosuppressed mice. After oral administration, cells from the Peyer’s patches released more IL-5, IL-6, and IgA. β-glucans increased IL-6 secretion of Peyer’s patch dendritic cells. Intestinal IgA production also increased after oral treatment. This study showed that β-glucans may activate Peyer’s patch dendritic cells and stimulate IL-6 and IgA production [176].

Other studies have shown that β-glucans boost infection protection by training or priming macrophages. When macrophages primed with β-glucans were administered into naïve mice, the macrophages developed an antimicrobial phenotype with higher levels of phagocytosis and ROS generation, as well as increased glycolytic and oxidative metabolism, mitochondrial mass, and membrane Potential. β-glucans produced extensive transcriptome changes in macrophages, as is consistent with early inflammatory response activation, followed by persistent modifications in metabolism, cellular differentiation, and antimicrobial function transcripts. The trained macrophages released proinflammatory cytokines and chemokines in response to LPS. Training was independent on Dectin-1 and TLR-2 expression [177].

The microalga *Euglena gracilis* contains vitamins, minerals, amino acids, and fatty acids. Paramylon is an insoluble fiber that is specifically found in *E. gracillis* with a triple-helical structure made of straight-chain 1,3 linked β-glucans. Paramylon modulates immunological function similarly to forms of β-glucan, demonstrating that branching is not a requirement for immune interaction [162,163].

Upper respiratory tract infections (URTIs) have been shown to be less severe when treated with paramylon compared to placebo. Paramylon has been shown to inhibit the growth of atopic dermatitis in mice, have anti-allergy effects, activate leukocytes, increase ROS production in neutrophils and monocytes, and activate the IL-1-mediated inflammatory response in human primary macrophages [178,179,180,181].

Murphy and Rezoagli et al. (2020, 2022) [128,182] demonstrated that β-glucan isolated from different species of mushrooms has the potential to reduce inflammation in lung injury in vitro. In this study, the authors used THP-1 macrophages, PMBCs from healthy donors, and lung epithelial cell lines to understand the effects of this sugar on both immune cells and epithelial cells of the lung. The authors demonstrated that different mushrooms have different β-glucan content. For example, *Lentinus edodes* had 70% *w*/*w* β-glucan content, while *Hypsizygus tessellatus* had 80% *w*/*w* β-glucan content.

These results also showed that, at 1 mg/mL, the extracts induced the inflammatory cytokines IL-6, TNF-α, IL-22, IL-8, IL-8, and TNF-α. However, after cytokine insulted the extracts, this reduced IL-8 and TNF-α in A549 and BEAS-2B lung cells. This work demonstrated that, in the absence of injury, β-glucans appear to prime cells. However, in the presence of injury, β-glucans have a more modulatory effect. This study also demonstrated that, although β-glucans are found in abundance in mushrooms and have the ability to reduce inflammation in vitro, the effects will vary between species [128].

In another study by the same authors, it was also shown that activity is also dependent on the extraction procedure [183]. In this study, lentinan was extracted from the *Lentinus edodes* by using an in-house procedure (IHL) and compared to a commercial extract. The results demonstrated that commercial lentinan (CL) had higher levels of α-glucans and less β-glucan compared to the in-house extract. In human alveolar epithelial A549 cells, both lentinan extracts decreased cytokine-induced NF-κB activation; however, the IHL extract was more efficient at lower concentrations. In contrast, the CL extract more efficiently reduced the production of pro-inflammatory cytokines-TNF-α, IL-8, IL-2, IL-6 and IL-22 [183].

The emergence of antibiotic-resistant pathogens and the use of vaccines to prevent and treat diseases, including cancer and chronic disorders, requires safe and effective adjuvants. However, few adjuvants are approved for human use, and none for mucosal administration due to potential toxicity and autoimmune diseases. Carbohydrate-based adjuvants, such as β-glucans, are a promising alternative due to their immunomodulatory properties, low cost, high tolerability, and potential for mucosal administration. These compounds activate specific receptors on monocytes and other cells, leading to the production of pro-inflammatory cytokines and immune enhancement. Moreover, α-glucans can also be chemically modified to increase their immunostimulant capabilities and used in particulate forms suitable for mucosal administration. However, further evidence is needed to ensure their safety and efficacy as adjuvants. Moreno-Mendieeta et al. (2017) provide a concise review on the potential of glucans as vaccine adjuvants [184].

In conclusion, β-glucans are natural compounds found in cereals and non-cereal sources such as microalgae, mushrooms, and yeast. They have several health benefits, such as improving gut health, reducing cardiovascular diseases and diabetes, enhancing the immune system, and having anti-inflammatory properties. Studies have shown that they can modulate immunological function similarly to forms of β-glucan, demonstrating that branching is not a requirement for immune interaction. β-glucans isolated from different species of mushrooms have the potential to reduce inflammation in lung injury in vitro, but their effects vary between species and depend on the extraction procedure used.

### 5.2. Fucoidan

Fucoidan, first isolated in 1913 [185], is a naturally occurring sulphated polysaccharide isolated from the extracellular matrix of brown seaweeds [86,87]. The polymeric composition of fucoidan is complex. Different species of brown algae contain fucoidan with different structures. The monosaccharide composition of the polysaccharide varies between species. The polysaccharide is mainly composed of L-fucose sugar and sulfate ester groups. This monosaccharide may bind to galactose (glactofucan), rhamnose (rhamnofucan), or both galactose and rhamnose (rhamnoglactofucan). The monosaccharide chain can also be composed of mannose, glucose, xylose, and arabinose [186]. This diverse composition influences biological activity [86,87]. Due to their heterogeneous composition, fucoidans can have various molecular weights. Low molecular weight molecules range up to 40 kDa, intermediate molecular weight ranges up to 140kDa and high molecular weight fucoidan ranges up to 330 kDa [146].

Fucoidan is one of the most prevalent polysaccharides found in brown algae and is isolated from the cell wall of brown seaweeds. Alginate, fucoidans, cellulose, hemicellulose, and other polysaccharides, as well as proteins, polyphenols, and other substances, make up the bulk of this structure’s complex matrix [187].

The polysaccharide can also be found in marine invertebrates such as sea urchins and sea cucumbers. Fucoidan derived from these sources is usually a linear polysaccharide; however, there is variance between sources. Fucoidans from *Acaudina molpadioides* and *Stichhopus horrens* are linear polysaccharides that are α 1→3 linked. Fucoidan from *A. molpadioides* consists of tetrafucose units. *S. horrens* fucoidan consists of monosaccharide units, and fucoidans from *Apostichopus Japonocas* consist of branched chains with repeating pentasaccharide units. Fucoidan from *Holothuria fuscopunctata* is composed of 1–4 linked monosaccharides [188,189]

Fucoidan has also been isolated from *Ferula hermonis* roots. This fucoidan fraction comprises fucose, glucose, sulphate, galactose, mannose, and certain certain proteins [190]. A fucoidan extract with antioxidant activity has also been isolated from the Eucalyptus plant [191].

#### Immuno-Modulatory Activity of Fucoidan

Fucoidan has been demonstrated to have several bioactive properties including anti-cancer, anti-microbial, anti-viral and anti-inflammatory activity [146,192,193,194,195]. Activity is strongly correlated to structure, with some activity dependent on sulphate content [196]. Extensive research has also been conducted on the beneficial effects of fucoidans in treating inflammatory diseases such as pancreatitis, colitis, osteoarthritis, and skin inflammation, as well as in addressing neurodegenerative diseases, immune dysfunction, and tumors. Fucoidans have also been found to be effective in treating a variety of health conditions, including diabetes, hepatic steatosis (fatty liver), liver fibrosis, renal ischemia, disrupted blood coagulation status, stem cell therapies, gastric ulcers, gout, bacterial and viral infections, and snake bites [197].

In their study, Kasai et al., (2015) observed that the anti-cancer action was dependent not only on the sulphate level but also on the location of the sulphate group on the sugar backbone [198].

Other studies have shown a correlation between activity and uronic acid or acetylated groups [199]. Acetylated fucoidan has been shown to induce macrophage receptors through recognition of glycoprotein receptors on the macrophage surface. These receptors included toll-like receptor-4 (TLR-4), scavenger receptor class A (SRA) and cluster of differentiation -14 (CD-14) [200].

The chemical composition of fucoidans affects their biological activity, including anti-inflammatory and immunomodulatory mechanisms. *Undaria pinnatifida* fucoidan increases IFN-γ levels and does not significantly affect IL-4, IL-6, TNF-α, or NF-κB levels. *Cladosiphon okamuranus* Tokida fucoidan has higher uronic acid content, leading to decreased IL-6 levels and attenuated NF-κB signaling. *Ascophyllum nodosum* fucoidan, which is highly branched, increases IL-6, IL-8, and TNF-α production from neutrophils, and *Turbinaria decurrens* polysaccharide reduces COX-2, IL-1β, and NF-κB signaling gene expression [197].

Fucoidan with high uronic acid content had a greater influence on immune cells, suggesting that immuno-modulating activity is correlated to the content of this component. This was determined by comparing fucoidan from *Ascophyllum nodosum* with a high uronic acid content to a version from *Fucus vesiculosus* with lower content [199].

Experiments found that fucoidan reduced the risk of tumor growth in in vivo models. Rats were used to examine fucoidan’s immunomodulatory effects against DMBA-induced mammary carcinogenesis. Oral administration of 200 and 400 mg/kg of fucoidan to rats was tested. Tumor incidence rates were reduced, tumor masses were diminished, and tumor latency was extended in fucoidan-treated groups. The production of IL-6, IL-12, and interferon gamma was boosted by fucoidan compared to the control group [201].

Another investigation using in vivo and in vitro models of lung cancer in A549 cells revealed that fucoidan might inhibit the expression of TGF receptors, thereby reducing cancer cell proliferation and progression [202].

Zhang et al. (2015) investigated the immune-modulatory effects of fucoidans derived from the species *Ascophyllum nodosum, Macrocystis pyrifera*, *Undaria pinnatifida*, and *Fuccus vesiculosus.* The effects on human neutrophil apoptosis, mouse NK cells, dendritic cells, and T-cells activation were measured. The adjuvant effect of these polysaccharides on antigen-specific immune responses was also measured. The results demonstrated that all fucoidan extracts inhibited apoptotic cell death by reducing the percentage of Annexin V+PI- cells which are in the early stage of apoptosis. Some of the extracts displayed a dose-dependent inhibitory effect on neutrophil apoptosis, with concentrations ranging from 5 to 100 μg/mL. There was a difference between species of fucoidan origin in terms of the concentration required for the effect. Fucoidan from *M. pyrifera* and *U. pinnatifida* displayed this potential between concentrations of 5 and 100 μg/mL, whereas *A. nodosum* and *F. vesiculosus* displayed positive results at 50–100 μg/mL (W. Zhang et al., 2015).

The production of pro-inflammatory cytokines from neutrophils was induced by all extracts. This included IL-6, IL-8, and TNF-α. *M. pyrifera* or *U. pinnatifida* induced the highest levels of cytokine production. When 50 mg/kg of *U. pinnatifida* and *F. vesiculosus* was injected intraperitoneally (I.P) into in vivo mouse models, the numbers of NK cells increased. This indicated that systemic administration of fucoidan had the ability to influence NK cells. The other fucoidans had no effect on this measured parameter.

When the fucoidans were injected intravenously, there was a significant decrease in the amount and proportion of spleen dendritic cells. All fucoidans, except for *U. pinnatifida*, increased mRNA levels of IL-6, IL-12p40, and TNF-α in splenocytes. The serum levels of IL-6, IL-12p70, and TNF-α were also increased. When compared to PBS therapy, all fucoidan administrations resulted in noticeably higher percentages of splenic CD4 and CD8 T cells that generated IFN-γ and TNF-α, the cytokines that distinguish Th1 and Tc1 cells, respectively, after 20 mg/kg I.P. administration. According to these findings, fucoidans may serve as an adjuvant by encouraging Th1-type immune responses. These results also demonstrated that fucoidan from *M. pyrifera,* compared to the other three fucoidans, has the greatest immune-activating impact on murine NK cells, DCs, and T cells, as well as on human neutrophils (W. Zhang et al., 2015).

To eliminate tumor cells, tumor vaccines aim to trigger cytotoxic T-cell responses. There is a very close relationship between cancer and the immune response. The inflammatory response may also influence the efficacy of therapies [203,204].

In-vivo treatment of fucoidan increases T cell activation and spleen cDC maturation, according to a study by Jin et al. (2014). When fucoidan was administered systemically, the expression of CD40, CD80, and CD86 as well as the production of IL-6, IL-12, and TNF- in spleen cDCs was increased. Additionally, in an IL-12-dependent way, fucoidan encouraged the development of IFN-producing Th1 and Tc1 cells. Fucoidan aided in the promotion of ovalbumin-specific antibody synthesis and stimulated IFN-γ production in ovalbumin-specific T cells. Additionally, fucoidan boosted ovalbumin-induced MHC class I and II up-regulation on spleen cDCs. Mice were also protected from the challenge with B16-OVA tumor cells by OVA vaccination with fucoidan as an adjuvant. Together, these findings imply that fucoidan may operate as an adjuvant to trigger a Th1 immune response and CTL activation, which may be beneficial for the development of tumor vaccines [144].

Clinical studies have shown interesting results for the use of fucoidan. A study by Takahashi et al., (2018) determined that fucoidan could be used to alleviate inflammatory conditions and improve the quality of life in patients with advanced cancer. This is an important study as there is a strong demand for anti-inflammatory medications for advanced cancers. The results showed that when patients were administered dry commercial fucoidan the levels of pro-inflammatory cytokines were decreased in the first two weeks after administration. Despite the lack of a control group and the small cohort, the findings are encouraging for using fucoidan to treat inflammation in advanced malignancies (Takahashi et al., 201). A Phase II clinical trial is currently recruiting for the administration of fucoidan as a supplemented therapy in metastatic colorectal cancer (Florean et al., 202). Patients with metastatic colorectal cancer were recruited for a randomized, double-blind investigation evaluating the effectiveness of low molecular weight (LMW) fucoidan as a dietary supplement. Fifty-four patients were included in the trial, and the findings demonstrated a statistically significant increase in the disease control rate (DCR), from 69.2% to 92.8%, between the study and control groups [205].

Fucoidan is a natural compound found in seaweed that has several bioactive properties. It has been found to have anticancer, antimicrobial, antiviral, and anti-inflammatory activities. The activity of fucoidan is correlated with its chemical composition, including sulfate content, uronic acid, and acetylated groups. Different types of fucoidan affect the immune system in different ways, and high uronic acid content has been found to have a greater influence on immune cells. Fucoidan has been found to have beneficial effects in treating various health conditions, such as inflammatory diseases, neurodegenerative diseases, and tumors. Research also suggests that fucoidan reduces the risk of tumor growth in in vivo models, inhibits the expression of TGF receptors, and has immune-modulatory effects.

### 5.3. Glucomannan

For millennia, traditional Chinese medicine has employed glucomannan to treat conditions such as asthma, cough, hernia, breast soreness, burns, and skin problems (Kumoro et al., 201).

The water-soluble polysaccharide glucomannan is considered a dietary fiber. It is a non-ionic linear polysaccharide made of acetyl-substituted 1,4-linked-b-D-mannopyranose and b-D-glucopyranose, with branches every 68 monosaccharides. The 3- and 4-monosaccharide branch chains are connected to the C-3 hydroxyl of glucose or the C-6 hydroxyl of glucose/mannose on the main chain. The mannose-to-glucose ratio is 1.6:1, and monosaccharide residues may be randomly distributed. The molecular weight ranges from 200 to 2000 kDa, dependent on origin and extraction techniques [206].

Glucomannan is a component of hemicellulose found in the cell walls of several plant species and microorganisms [207]. It can be obtained from numerous botanical sources, but porang (*Amorphophallus oncophyllus*) tuber is the most viable due to its high glucomannan content and sustainable supply. Porang is a perennial plant that is indigenous to the Indonesian jungle, where it is widely cultivated. Its root is a significant source of glucomannan [208] containing 8–10% of this polysaccharide. Other sources include *Bletilla striata* (Wang et al., 201).

#### Immuno-Modulatory Activity of Glucomannan

Mannans have been shown in clinical studies to exhibit vaccine-adjuvant qualities, which are most likely mediated by their interaction with mannose receptors. It has been demonstrated that oxidized mannan mucin-1 can be used as an adjuvant in breast cancer immunotherapy. A 12-to-15-year follow-up of a clinical trial revealed that this compound lowers cancer recurrence rates and increases recurrence intervals, without causing toxicity or adverse responses. The trial indicated that targeting the mannose receptor on macrophages and dendritic cells results in robust cellular immune responses [209].

Many human clinical studies have been undertaken to demonstrate the efficacy of oxidized mannan-MUC1 fusion protein (M-FP) as an anticancer vaccine in MUC1+ adenocarcinoma patients. The recurrence incidence in patients receiving placebo was 60% after 12–15 years of follow-up (9 of 15). The incidence was 12.5% in individuals undergoing immunotherapy (M-FP) (2 of 16). The recurrence time in the placebo group varied from 7 to 180 months, while the recurrence time in the vaccination group was 95 and 141 months (mean: 118 months) following surgery [209].

During the 12–15-year follow-up, no patients who had M-FP injections exhibited any harmful effects or symptoms of autoimmunity. Preliminary evidence suggests that M-FP improves overall survival in patients with early stage breast cancer [209].

*Dendrobium officinale* glucomannan extract exhibited no toxic effect on in vitro macrophages. The extract induced a dose-dependent NF-κB phosphorylation and IκBα degradation. The extract activated NF-κB signaling without IκBα. Anti-mouse TLR4 monoclonal antibodies suppressed glucomannan-induced NF-B activation in THP-1 cells, indicating it activates the immune response through TLR4. RT-PCR was used to evaluate the expression of NF-κB genes. When NF-κB phosphorylation inhibitors were introduced, gene expression was normal. O-acetylated glucomannan elicited an immunological response through NF-κB and TLR4. This study also demonstrated that CL4 and IP10 are new immune response targets of O-acetylated glucomannan.

As an alkylating anticancer medication, cyclophosphamide (CTX) causes bone marrow suppression, immunosuppression, oxidative stress, and other adverse effects. It is used to treat systemic lupus erythematosus, lymphoma, and autoimmune illnesses. CTX treatment damages normal cells, reducing body weight, spleen and thymus weights, leukocyte and NK cell activity, and immune organ function, causing secondary infections. Glucomannan can be used to attenuate CTX-induced immunosuppression. A study evaluated glucomannan from *Amorphophallus konjac* on Cyclophosphamide-induced immunosuppressed mice [159].

The results showed that glucomannan reduces spleen and thymus indices and weight loss. These studies showed that the extract could recover the immune effect of CTX treatment. CTX therapy decreased splenic lymphocyte proliferation considerably compared to control. The glucomannan promoted superior B lymphocyte proliferation at low doses compared to CTX. Hemolysin is an indicator of humoral immunity. Treatment lowered serum hemolysin levels to raise antibody levels, indicating that the extract might boost and control humoral immune activity through serum hemolysin. The treatment improved macrophages’ capacity to stimulate the innate immune response against foreign molecules and was stable in vivo. Treatment also reduces IL-2, IgG, and TNF-α production.

Thus, mannans have vaccine-adjuvant qualities mediated by their interaction with mannose receptors. Oxidized mannan mucin-1 can be used as an adjuvant in breast cancer immunotherapy, which lowers cancer recurrence rates and increases recurrence intervals, without causing toxicity or adverse responses. Human clinical studies have demonstrated the efficacy of oxidized mannan-MUC1 fusion protein (M-FP) as an anticancer vaccine. Dendrobium officinale glucomannan extract activates NF-κB signaling without IκBα and elicits an immunological response through NF-κB and TLR4. Glucomannan can be used to attenuate CTX-induced immunosuppression by reducing spleen and thymus indices and weight loss, improving macrophages’ capacity to stimulate the innate immune response and raising antibody levels.

### 5.4. Chitosan

Chitosan is a polysaccharide derived from chitin [99]. Chitin is a natural polymer with repeating units of *N*-acetyl-d-glucosamine [100]. Chitosan occurs naturally in cell walls of some fungi [100]. However, chitosan is most obtained industrially by deacetylation of chitin extracted from a range of eukaryotic species such as crustacea, insects and fungi [99,100]. The common sources most cited in literature as raw material for chitosan preparation are shrimp and crabs, while other species such as lobster, crayfish, krill and oyster have also been utilized [99].

When subjected to deacetylation, the repeating units are predominantly without the acetyl functional group, i.e., d-glucosamine, and linked together by a glycosidic β-(1-4) bond making up chitosan [99,210]. Its molecular structure has an amino group (C2) and two hydroxyl groups (C3 and C6) which form intermolecular hydrogen bonds that determine the stability of the polymer [100].

The molecular fraction of the *N*-acetylated repeating units is defined as the degree of acetylation (DA), while the percentage of the repeating units of β-1,4-d-glucosamine in the polysaccharide is defined as the degree of deacetylation (DD) [99]. Most commercial chitosan has DD values between 70 and 90%. The molecular weight (MW) of this natural polymer ranges from 10,000 to 1 million Daltons [100]. As reported in numerous studies, DD and MW are the essential parameters that influence the bioactivity of chitosan [99].

Chitosan is known as a bioactive compound with numerous biological properties, such as antitumoral, immunoenhancing, antifungal, antioxidant, antihypertensive, anti-inflammatory, anticoagulant, antitumoral, antimicrobial, hypocholesterolemia, antidiabetic, and wound-healing activities [100,210]. Its diverse biomedical attributes, together with exceptional properties such as non-toxicity, biodegradability, biocompatibility, non-antigenicity, and low cost, have led to extensive pharmaceutical applications. This includes drug delivery systems, tissue engineering, food technology, bioimaging, implants, contact lenses, gene delivery, and protein binding [210].

Drug delivery systems based on polysaccharides have revolutionized medical treatments due to their efficiency and specificity. Among various polymers, chitosan-based drug delivery systems are attracting substantial interest as vehicles that can release their contents at the desired rate and location in the body [210].

#### Immuno-Modulatory Activity of Chitosan

Early studies demonstrated that chitosan can activate macrophages [211] and induce cytokine secretion from natural killer (NK) cells [212]. These effects have been shown to occur in a phagocytosis-dependent manner.

Recently, it was reported that chitosan promotes dendritic cell maturation by inducing production of type I interferons (IFNs) and enhancing antigen-specific T-helper 1 (Th1) responses in a type I IFN receptor-dependent manner. Further findings from this study demonstrated that chitosan can engage the STING-cGAS pathway to trigger innate and adaptive immune response [213]. Chitosan has also been shown to promote inflammasome activation [214].

Low-molecular-weight chitosans (LMWCs) were tested for immunostimulatory properties in RAW264.7 macrophages. Two LMWCs (3 kDa and 50 kDa) had immunostimulatory activity that was dosage-dependent and, at higher doses, molecular-weight-dependent. LMWCs increased pinocytic activity and induced the production of TNF-α, IL-6, IFN-, NO, and iNOS in a molecular-weight- and concentration-dependent manner. LMWCs enhance iNOS and TNF- gene expression. In a molecular-weight- and concentration-dependent way, LMWCs upregulated proinflammatory cytokine mRNA expression and activated RAW264.7 macrophages [151].

Chitin and chitosan are also capable of boosting antiviral immune responses through the activation of innate immune cells. It has been demonstrated that they can increase the number of phagocytes and that the N-acetyl-glucosamine residues in chitosan can increase the production of reactive oxygen species (ROS), the secretion of nitric oxide (NO), and myeloperoxidase activity in phagocytes. In addition, these polymers can stimulate neutrophil migration and humoral responses at the systemic (IgG) and mucosal (IgA) levels [215].

To understand the protective effect of chitin as an adjuvant, BALB/c mice were infected with Leishmania and given chitin microparticles (CMPs) as an adjuvant. The mice were infected three weeks after the first inoculation. The results demonstrated that the group that was administered leishmania antigen and CMP had stronger IFN-γ and IL-10 in spleen cell culture and lower IgG1 in sera than the control [152].

Chitosan has been shown to activate macrophages and induce cytokine secretion from NK cells, while also promoting dendritic cell maturation, inflammasome activation, and immune responses via the STING-cGAS pathway. Low-molecular-weight chitosan has immunostimulatory activity in macrophages and upregulates cytokine expression. Chitin and chitosan can boost antiviral immune responses by increasing phagocyte numbers and stimulating neutrophil migration and humoral responses. Chitin microparticles have a protective effect as an adjuvant against Leishmania in mice by inducing stronger IFN-γ and IL-10 and lower IgG1.

### 5.5. Alginate

Alginate is a linear polymer composed of two kinds of uronic acids: β-d-mannuronic acid (M units) and α-l-guluronic acid (G units) [216,217]. The two uronic acids are bound in block-wise segments in the form of repeated G units (GG), repeated M units (MM), and alternating G and M units (GM), linked via (1→4) glycosidic bond [101,102,103,216]. The chains are composed of a random sequence of M- and G-blocks that are interspersed with regions of alternating MG blocks [218].

The varying block sequence in an alginate chain depends on algal source (species), geographical location, harvest season, and extraction method [101,102,103]. The six-membered sugar rings are rigid and restrict rotation around the glycosidic bonds, giving alginate the character of stiff molecules [218]. The electrostatic repulsion between the charged groups distributed along the polymer chain further contributes to the rigidity of the molecule [218].

Therefore, the physical and mechanical properties of alginate depend strongly on the proportion of the M and G units and the block sequence in the chain [103,218]. The molecular weight of alginate ranges between 32,000 and 400,000 g/mol [102] Alginate is also considered a polyelectrolyte and can be characterized as an anionic copolymer due to its abundance of free hydroxyl (OH^-^) and carboxyl (COO^-^) groups distributed along the polymer chain backbone [219].

Unlike neutral polysaccharides, alginate has two types of functional groups that are available and can be modified to alter the characteristics in comparison to the parent compounds [216,219].

Alginate is Generally Recognized as Safe (GRAS), nontoxic, satisfactorily biocompatible, and sufficiently biodegradable [216]. Industrial applications of alginates are found mainly in food, medical, cosmetic, pharmaceutical, and textile industries and are linked to their physiochemical properties [216,218].

The ability of alginate to retain water, as well as its gelling, viscosifying, and stabilizing properties, is the major reason for its use in soft tissue engineering and wound healing [102,218]. Nowadays, alginates are typically mixed or modified to suit other types of biomedical applications [102]. There have also been claims that alginate with high M content tends to be more immunogenic (able to produce an immune response) than alginate with high G content [102].

Alginate is a natural polysaccharide that occurs in the structural component of brown marine algae (*phaeophyceans*, also called brown seaweeds) and some from soil bacteria such as *Pseudomonas aeruginosa* [216,218]. Alginate constitutes a key component of the seaweed cell walls and also appears to be present in the intercellular space matrix [101].

The presence of alginate provides flexibility and a strong structure to the algae and protects them to a certain extent from injury when the algae are exposed to strong sea waves [102]. In bacteria, it forms a protective capsule, aids in biofilm formation, and assists in bacterial adherence and colonization [102]. Commercial alginate is extracted from brown seaweed, primarily from *Laminaria*, *Macrocystis*, *Ascophyllum, Sargassum*, and *Fucales* species [101,102]. In *Laminaria* and *Macrocystis* species, alginate comprises up to 40% of the dry matter. However, it has been shown that different species and extraction techniques yield different amounts and quality.

#### Immuno-Modulatory Activity of Alginate

To date, there are not many studies reporting the immunomodulatory effects of alginate polymer. Due to its reliable structural stability and physicochemical properties, it is mainly used in drug delivery systems as a gelling agent, polymer matrix, and coating and in drug encapsulation [220].

However, alginate is known as an immunostimulant in shrimps. A study conducted by Yudiati et al. (2019) performed experiments to evaluate dose-dependent effects of sodium alginate on immune parameters, immune-related gene expression, and white spot syndrome virus (WSSV) resistance in Pacific white shrimps. It was found from this study that sodium alginate was able to stimulate the innate immune system in Pacific white shrimp, as there was an increase of non-specific immune parameters and upregulation of immune-related genes. The findings of this study were also able to illustrate the interaction among the cellular and humoral substances, the expression of three immune-related genes, and the resistance against WSSV infection [221].

There have been reports of clinical studies and applications of the patented (DE-102016113017.6) G2013 molecule, which is the α-l-guluronic acid monomer. It has been stated in various studies that G2013 is a novel drug with immunomodulatory property and could be classified as a non-steroidal anti-inflammatory drug (NSAID) [222,223,224,225]. Furthermore, with its low molecular weight, this compound has lower toxicity against GI tract and kidney function compared to other NSAIDs [225].

Bi et al. (2017) studied the effect of alginate on macrophage phagocytosis in murine RAW264.7 cells. Alginate can accelerate intracellular phagocytosis of gold nanoparticles, fluorescent microspheres, and IgG-opsonized *Staphylococcus aureus*. Alginate enhanced the expression of the TLR4 receptor and activated Akt/NF-B and p38 MAPK signaling pathways. TLR4, NF-B, and p38 MAPK inhibitors and TLR4 gene knockdown inhibited alginate-induced phagocytosis, demonstrating their participation. This is the first study to suggest that alginate activates macrophages by upregulating TLR4 expression and boosting the Akt/NF-B and p38 MAPK signaling pathways [153].

Although not much is known about its applicability for human use, hopefully the numerous studies and reports will show the potential immunomodulatory effects of alginate and adapt it to further human clinical studies.

### 5.6. Hyaluronic Acid

Hyaluronic acid (HA) is a natural polysaccharide belonging to a class of mucopolysaccharides called glycosaminoglycans. Glycosaminoglycans are long linear polysaccharides which consist of repeating disaccharides, typically uronic sugar, and an amino sugar [226]. Specifically, HA is a simple glycosaminoglycan consisting of two sugars, D-glucoronic acid and N-acetyl-D-glucosamine linked by β (1, 4) and β (1, 3) glycosidic bonds [227]. HA is ubiquitous in humans and is also produced by some bacteria [228], either naturally or through biotechnological processes. This allows for ease of accessibility for a variety of applications.

HA lends itself to a wide range of applications due to its biocompatibility and biodegradability, which have been well demonstrated [229,230]. The function of HA is dependent on the molecular weight of the polymer. In its native high-molecular-weight form, it is used primarily as a lubricating agent for joints in conditions such as osteoarthritis [231], for ophthalmology applications, or for topical wound healing [232]. However, as this is an endogenous polymer, it can be easily broken down by the body’s natural defenses. HA-specific enzymes are known as hyaluronidases. These enzymes break down HA from a large-molecular-weight polymer in excess of 1000 kDa to smaller fragments, which illicit various effects which are molecular-weight dependent. The effects of HA in the body vary greatly with molecular weight, as evidenced by the anti-inflammatory effects of high-molecular-weight (>1000 kDa) and the pro-inflammatory effects of low-molecular-weight (<200 kDa) HA [233].

To counter the uncertainty related to molecular weight and the associated effects due to enzymatic degradation of HA, this polysaccharide can be manipulated through any of its various functional groups to yield a modified HA, which is more resistant to enzymatic breakdown. HA can be modified via the carboxyl, hydroxyl, or amino functional groups [234]. The most utilized functional groups are carboxyl and hydroxyl because of their relative reactivity. Through these functional groups, HA can be modified with relative ease to possess groups such as thiols or methacrylates, which can be further reacted to produce highly crosslinked structures. This type of click-chemistry photo-coupling involving thiols has become one of the most promising strategies for the preparation of hydrogels due to the high specificity, yield, bio-orthogonality, and mild reaction conditions [235].

HA is found throughout the human body and vertebrate tissues as a primary constituent of the extracellular matrix, playing roles in cellular proliferation, migration, and differentiation. Additionally, it is found at different concentrations and molecular weights in the umbilical cord, synovial fluid, dermis, epidermis, thoracic lymph, and urine [236].

HA is one of the only mucopolysaccharides not produced by the Golgi apparatus [237]. In vertebrates, HA is synthesized via three different hyaluronan synthases (has)—has1, has2, and has3. These isozymes function to lengthen the polysaccharide by the repeated addition of glucuronic acid and N-acetyl-D-glucosamine, which are then extruded through the cell wall via ABC transporters. The different forms of proteins possess other kinetic profiles, which ultimately affect the size of the HA produced [238]. Has1 and has2 proteins are moderately active and implicated in the synthesis of high-molecular-weight HA, whereas has3 proteins are highly active and produce low-molecular-weight HA [239].

In bacteria, such as the *Streptococcus* genus, the genes required to synthesize HA are known as *hasA*, *hasB*, and *hasC*. The Streptococcus genus has evolved to produce HA endogenously; however, they are renowned to produce several endotoxins which would render the HA produced unsuitable for human or animal use and, therefore, unsuitable for industrial processes. To overcome this issue, strains which are generally regarded as safe (GRAS), such as *Lactococcus lactis*, are being researched and engineered.

#### Immuno-Modulatory Activity of Hyaluronic acid

HA is known for its unique physicochemical properties; however, few investigations of the importance of molecular weight in immunological response have been carried out. It has been demonstrated that high-molecular-weight HA, <1000 kDa, possesses antiangiogenic, anti-inflammatory, and immunosuppressant effects, whereas the converse is true of low- and moderate-weight HA [240]. Low-molecular-weight HA has been found to activate the NF-κB pathway,l which is associated with inflammation, whereas high-molecular-weight HA suppresses the production of lipopolysaccharide, a bacterial endotoxin [241]. These conflicting effects of molecular weights can be manipulated for a variety of applications.

One such application is the use of HA as a vaccine adjuvant to increase the immune response. Studies have shown that HA adjuvanted vaccines derived from *Staphylococcus aureus* produced a greater immune response than that observed with pure protein derivative adjuvanted vaccines [242].

Additionally, patients suffering from chronic obstructive pulmonary disease (COPD) were given high-molecular-weight HA in a pilot study of 41 patients. Lowered systemic inflammatory biomarkers and significantly shortened instances of non-invasive positive-pressure breathing were observed in HA-treated patients, revealing the anti-inflammatory potential of high-molecular-weight HA [243].

HA is a polymer, which has found various uses, from supplementation to cosmetics to tissue engineering. Most recently, the focus has shifted to the immunomodulatory properties of HA and the dependency of these properties on molecular weight.

### 5.7. Carrageenan

Carrageenan (CRG) is the main polysaccharide in the cell walls of red seaweeds and can constitute from 30 to 75% of the total algal dry weight [244]. These polysaccharides are anionic linear sulphated structures of high molecular weight that are formed by alternating α-1,3 and β-1,4 linkages in a shared backbone of D-galactose, having one (κ-CRG), two (ι-CRG), or three (λ-CRG) sulphates per disaccharide unit [112] Flexible gels can be obtained with κ and ι-CRGs in ionic solution, while λ-CRG is non-gelling, as it does not self-associate into helical structures. CRG has a wide application in the food, pharmaceutical, and cosmetics industries. Its thickening properties allow the direct incorporation into skin products such as creams, sunscreen, and soap [112].

#### Immuno-Modulatory Activity of Carrageenan

The physicochemical properties of carrageenan have also stimulated its use for drug delivery and tissue engineering. Chan et al. (2017) tested monocyte behavior in vitro when exposed to κ, ι, and λ-CRG. Only λ-type activated the immune system promoting monocyte adhesion and binding to IL-8. The human non-specific immune response can be modulated by CRG oligosaccharides, activating different interleukins and increasing macrophage phagocytosis, natural killer cell activation, and lymphocyte proliferation [245].

Yermak et al. (2020) reported the inhibitory effects of CRGs on LPS-induced inflammation in complex therapy for patients with enteric infections due to *Salmonella* and in a mouse model of endotoxemia. The synthesis of anti-inflammatory IL-10 was increased by the presence of CRGs. Low concentrations of CRGs in the mixture with LPS presented higher activity. When supplemented in patients, CRG improved biochemical indicators and immune parameters. In mice, it increased the non-specific resistance of mice to *E. coli* LPS [246].

Kalitnik et al. (2017) reported on the in vivo effect of carrageenan which was isolated from *Tichocarpus crinitus* on LPS-induced endotoxemia in mice. When orally administered at a dose of 100 mg/kg, it induced the secretion of IL-10 an anti-inflammatory cytokine. When animals were administered carrageenan as a prophylactic before LPS administration, IL-10 was increased 2.5-fold, and TNF-α, an inflammatory cytokine, was reduced 2-fold when compared to control groups. A pretreatment with carrageenan greatly decreased the activation of inflammatory cells [247].

Harikrishnan et al. (2021) reported the positive influence of a diet containing 20 g/kg of κ-CRG in *Rachycentron canadum* (cobia fish) against the emerging pathogen *Lactococcus garvieae.* The group treated with κ-CRG presented greater growth, improvement in anti-inflammatory cytokine and chemokine regulation, and in innate–adaptive immune performance [190].

In summary, Carrageenan has physicochemical properties that make it useful in drug delivery and tissue engineering. It can modulate the non-specific immune response and increase macrophage phagocytosis, natural killer cell activation, and lymphocyte proliferation. Carrageenan has also been found to have inhibitory effects on LPS-induced inflammation in complex therapy for patients with enteric infections due to Salmonella and in a mouse model of endotoxemia. Additionally, a diet containing κ-CRG was found to positively influence anti-inflammatory cytokine and chemokine regulation and innate-adaptive immune performance in Rachycentron canadum.

### 5.8. Xylans

Xylans are non-cellulosic polysaccharides [248]. Xylan from different sources can differ highly in structural complexity in terms of sugar constituents, glycosidic linkages, and the structure of the glycosyl side chains [114,249]. This structural diversity is related to their functionality in the plant [114]. Xylans consist of (1→3) and (1→4)-linked β-xylosyl linear backbones, generally known as homoxylan [248,249]. The xylosyl units may be substituted variously with short side chains containing l-arabinose, galactose and xylose (Xyl) and are present in grasses and cereals [248,249]. Higher plants, such as hardwoods, may have acetyl substituents at the C-2 and/or C-3 of the xylosyl unit, generally called heteroxylan [249]. The repeating unit of xylosyl that comprises the backbone of this biopolymer has two reactive hydroxyl groups that offer various possibilities for region-selective chemical and enzymatic modifications [114].

After cellulose, xylans are the second most abundant polysaccharides in nature. They are sourced from wood and algae, as well as various plants such as grasses, cereals, and herbs [114,250]. They are known to occur in several structural conformations. Sources of xylans include many agricultural crops, such as straw, sorghum, sugar cane, corn stalks and cobs, hulls and husks from starch production, and forest and pulping waste products from hardwoods in particular [114].

Plants that do not produce xylan have weak stems and cannot bear weight [251]. Xylans constitute 25–35% of the dry biomass of woody tissues of dicots and lignified tissues of monocots and occur at up to 50% in some tissues of cereal grains [114].

Typically, plant cell walls are divided into two categories of primary and secondary walls. Primary cell walls surround cells within the plant that are capable of growth or those that are actively growing. Secondary cell walls are usually described as thickened structures that contain lignin, polysaccharides, and hemicelluloses [252]. Thus, xylans are mostly deposited as a component of hemicelluloses (heterogeneous polymer composed of many sugars) in the secondary cell walls of plants [251].

#### Immuno-Modulatory Activity of Xylans

Xylans from natural sources have various biological activities, such as antioxidant, antitumor, and prebiotic activities, and have potential nutritional and pharmaceutical value in healthy foods and medicines. However, xylans from natural sources require a high concentration to exert their pharmacological effects and lack the efficacy of most conventional drugs [253]. Several studies have reported stimulatory activity of xylans either in vivo or in vitro, although not many studies were able to relate their structure with the reported stimulatory properties [254].

Only a few heteroxylans occur among the numerous polysaccharides exhibiting immunomodulatory and antitumor activities in various biological tests. Glucuronic-acid-containing (acidic) xylans isolated from annual plant residues such as bamboo leaves, corn stalks, wheat straw, and hardwood have been reported to significantly inhibit the growth of tumors, probably due to the indirect stimulation of the nonspecific immunological host defense. More immunostimulant effects have been reported for arabina-(glucurono) xylans isolated from numerous plant species [255].

A study conducted by Akhtar et al. reported that Arabinoxylans from wheat bran have the potential to stimulate the antibody-mediated immune response in chickens and could be explored as a low-cost alternative to allopathic drugs to prevent avian coccidiosis [256].

The structure/function relationship of two acidic heteroxylan types from corncobs and beechwood were studied, and it was found that neither the uronic acid content nor the distribution pattern of the uronic acid side chains was found to be a determinant for the expression of the immunomodulatory activity [255]. This finding agreed with the findings of Kardošová et al.: when compared to drugs used in clinical practice, the isolated xylan exhibited a much greater effect in treating cough in cats regardless of its uronic acid distribution on the polymer chain [257].

### 5.9. Ulvan

*Ulva* (sea lettuce) species are widely consumed as food. Besides the carbohydrate content, they are a source of high levels of nutrients, fatty acids, proteins, fiber, and tocopherol compounds, presenting high growth rates and easy adaptation to land-based aquaculture [113]. Ulvan is a cell wall polysaccharide that can represent from 9 to 36% of the dry weight of algal biomass, and it is the only polysaccharide from the four (ulvan, cellulose, glucuronan, and xyloglucan) present in the sea lettuce cell wall that contains both iduronic acid and rhamnose [114].

Ulvan is a class of water-soluble polysaccharide that is formed mainly by sulphated rhamnose, uronic acid, and xylose obtained from green algae, including *U. armoricana, U. fasciata, U. ridigida*, and *U. flaccam* [193].

#### Immuno-Modulatory Activity of Ulvan

Ulvan extracts have been extensively investigated by the pharmaceutical and agricultural sectors as alternative sources for obtention of a range of high-value products. Antibacterial, antiviral, anticoagulant, antioxidant, cytotoxic, antitumor, and immunostimulant properties are only a few of the activities reported for ulvan polysaccharides [60]. The sulphate content is usually from 14 to 23% in ulvan extracts and has been associated with immunomodulation and cell protection from oxidative stress. Modifications such as acetylation, phosphorylation, and benzoylation have been reported to increase the antioxidant activity of ulvan extracts [114].

The immunomodulatory activity of ulvan has been widely reported. Ref. [258] reported higher releases of nitric oxide, IL-6, IL-10, IL-12, TNF-α, and IL-1β cytokines when exposing murine macrophage cells (RAW264.7) to crude water-soluble sulphated polysaccharides extracted from *Ulva intestinalis* [259]. The authors also characterized the extract, which had a molecular weight of 8.71 × 10^4^ g/mol and neutral sugars (58.7%), sulfates (6.2%), proteins (3.2%), and uronic acid (4.9%).

In cancer cell lines, ulvan from *Ulva lactuca* showed a significant cytotoxic effect against Hela, HepG2, and MCF7. Rhamnose, galactose, xylose, glucose, mannose, uronic acid (21.5%), and sulfate (18.9%) were identified in the extract, with a molecular weight of 3.47 × 10^5^ g/mol [260]. Ulvan also positively affected human embryonic kidney 293 (HEK293) and porcine intestinal epithelial cells, where mRNA protein expression of cytokines increased in porcine cells, and ulvan extract stimulated TLR4 expression in HEK cells [261].

Most studies report the in vitro application of ulvan extracts. Fernandez-Diaz et al. (2017), however, incorporated ulvan from *Ulva ohnoi* into chitosan-based nanoparticles to analyze the immunostimulant effect in *Solea senegalensis*, a species of flatfish. This approach is important to facilitate the delivery and administration of extracts in aquaculture. The authors tested native extracts and fractions obtained by enzymatic and chemical reactions. Immune response was affected by ulvan variation according to chemical and enzymatic methods, and only native ulvan nanoencapsulation significantly increased the immune response in *S. senegalensis* macrophages.

Ulvan extracts from various types of algae have been studied for their potential pharmaceutical and agricultural applications. They possess antibacterial, antiviral, anticoagulant, antioxidant, cytotoxic, antitumor, and immunostimulant properties. Ulvan’s immunomodulatory properties have been extensively reported, and studies have shown that it can increase the release of cytokines, including nitric oxide and TNF-α. In vitro studies have demonstrated ulvan’s cytotoxic effects on cancer cell lines, and it has been shown to enhance the immune response in fish.

## 6. Conclusions

Immunobiological therapy treats illnesses by affecting the immune system, either activating or suppressing it. The way the substance is administered, and its ingredients are vital to the response, as the immune system is specific at detecting foreign materials.

Biomaterials have critical functions in healthcare, such as tissue engineering, drug delivery, and immunotherapies. Polymers, which are natural or synthetic compounds made of macromolecules, have various properties that make them useful, such as biocompatibility, biodegradability, and chemical modifiability. Polysaccharides, which are abundant natural polymers, and biomacromolecules have various therapeutic properties, such as anti-inflammatory and immunomodulatory effects. However, developing natural polysaccharides into therapeutics is complex because of their source, species, molecular weight, composition, and structure, which affect their efficacy.

Although natural polysaccharides are advantageous over synthetic ones because they are safe, inexpensive, stable, hydrophilic, and biodegradable, there are some disadvantages. Extraction and purification are costly and time-consuming, and the structure–activity relationships are not well-documented. Nonetheless, natural polysaccharides have vast potential in biomedical applications, as they have a strong influence over the immune system and can be tailored for specific purposes. Defining the structure–activity relationship between immune cells and natural polysaccharides is necessary for them to be utilized fully.

## Figures and Tables

**Figure 1 polymers-15-02373-f001:**
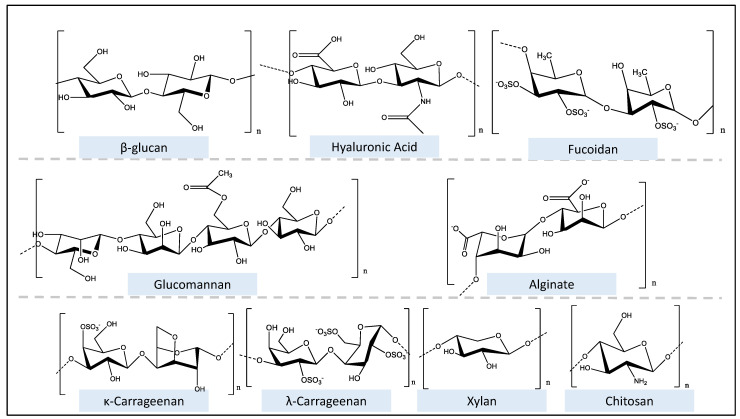
Chemical structures of naturally occurring polysaccharides.

**Figure 2 polymers-15-02373-f002:**
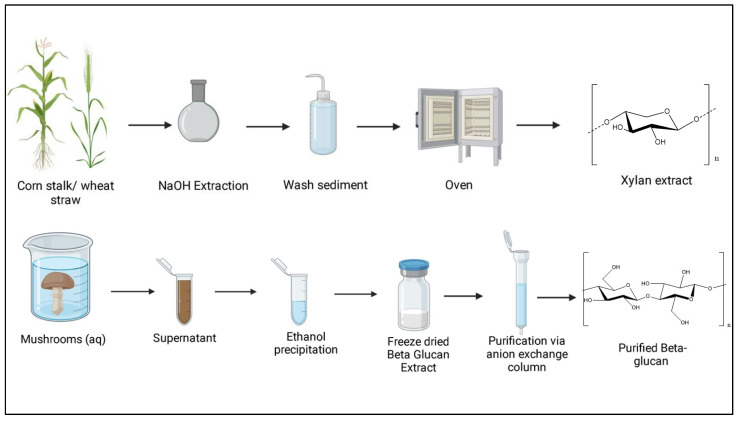
Graphical representation of methods used for the extraction of xylan from corn stalk/wheat straw and beta-glucans from mushroom sources.

**Figure 3 polymers-15-02373-f003:**
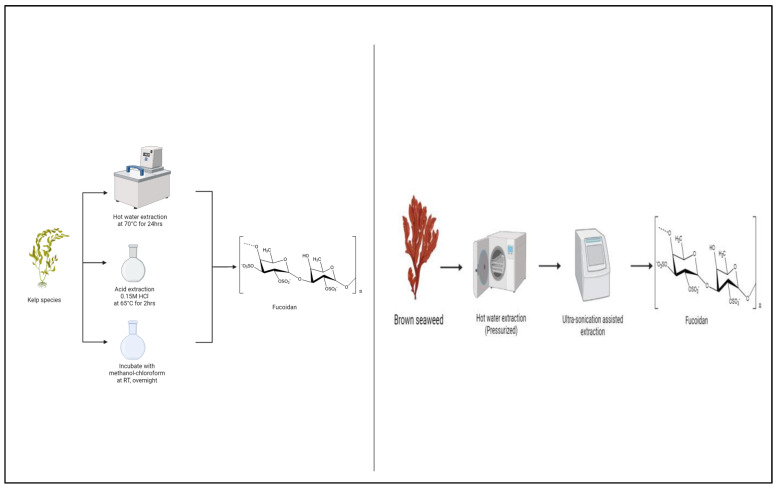
Graphical representation of methods used for the extraction of Fucoidan from kelp species and methods used for the extraction from brown seaweed.

**Figure 4 polymers-15-02373-f004:**
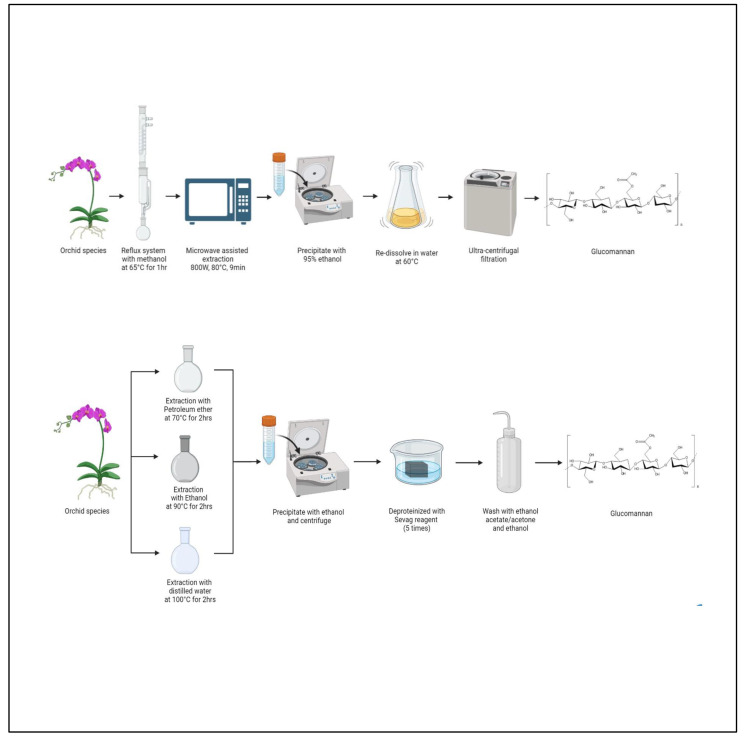
Graphical representation of two extraction methods of glucomannan from orchid species.

**Figure 5 polymers-15-02373-f005:**
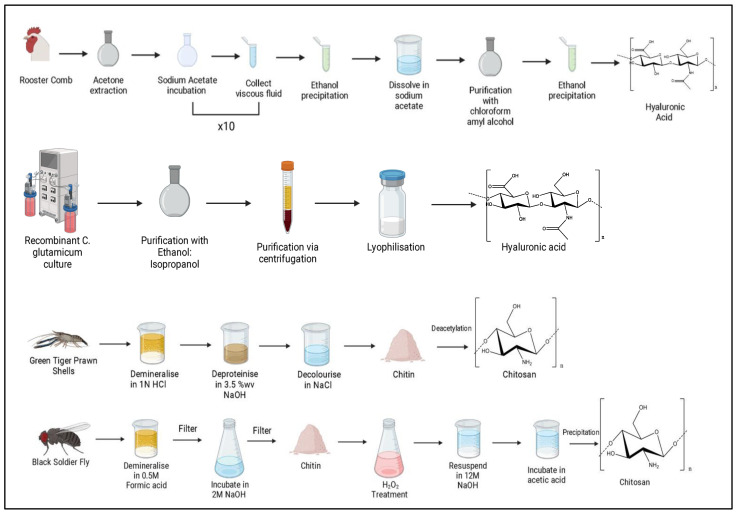
Graphical representation of methods used for the extraction of hyaluronic acid from *C. glutamicum* cultures and rooster comb. Methods used for the extraction of chitosan from green tiger prawn shells and black solider flies.

**Figure 6 polymers-15-02373-f006:**
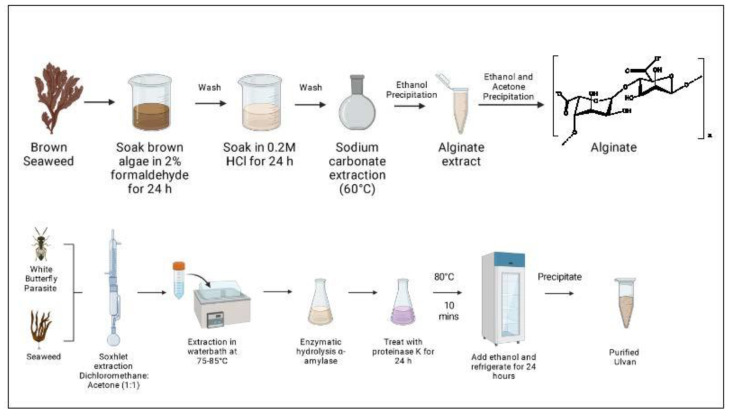
Graphical representation of methods used for the extraction of alginate from brown seaweed and ulvan from white butterfly parasite and seaweed.

**Figure 7 polymers-15-02373-f007:**
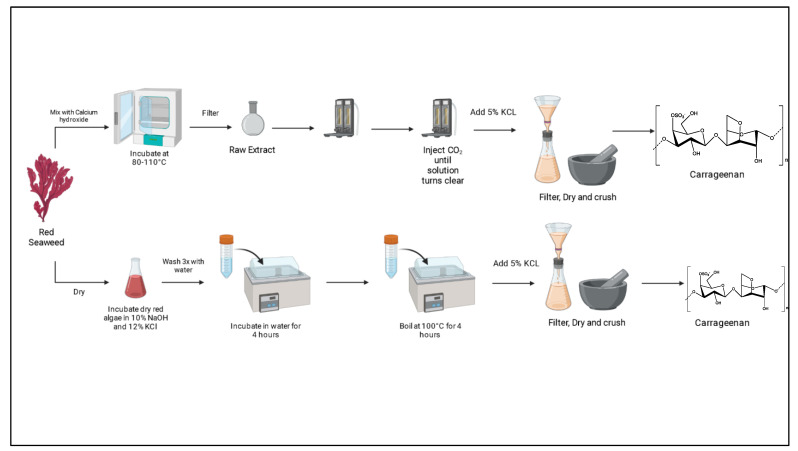
Graphical representation of methods used for the extraction of carrageenan from red seaweed.

**Table 1 polymers-15-02373-t001:** Methods of extraction for polysaccharides from natural sources. PSR, polysaccharide; DW, deionized water; DTW, distilled water; HA, hyaluronic acid; SDA, sodium acetate; DCM, dichloromethane; RT, room temperature; RMO, rhamnose; XYL, xylose; UA, uronic acid; MA, mannuronic acid; GA, guluronic acid; GLU, glucose; MAN, mannose.

**Polysaccharide**	**Species**	**Extraction Method (EM) and Purification (PFN)**	**Ref.**
**Beta-glucan** **(C_6_H_10_O_5_)_n_** **Up to 3000 kDa**	*Lentinus* *edodes* Mushroom	**EM** A water-extracted residue was used to extract the alkaline-soluble crude PSR. The supernatant was then neutralized, dialyzed, precipitated (1:4 *v*/*v*) in EtOH, and freeze-dried. **PFN** Extract diluted in 15 mL of DW and loaded onto a Q-Sepharose Fast Flow strong anion-exchange column (QFF, 4.6 15 cm). At a flow rate of 4 mL/min, the PSR was eluted with 600 mL of DTW and 0.2 and 0.4 mol/L NaCl solutions, sequentially.	[62]
	*Saccharomyces cerevisiae*	**EM** Yeast cultured in extract–glucose broth for 48 h, heated for 2 h at 80 °C, centrifuged, and then rinsed with DTW. Samples centrifuged and a five-fold CH_3_COOH extraction was used to extract β-glucan from lysed yeast pellets.	[63]
**Xylans** **(C_5_H_8_O_4_)_n_** **Up to 200 kDa**	Wheat straw and corn stalks	**EM** Hot alkali extraction. Raw materials were extracted in sealed with NaOH, and optimal conditions were 100 °C, 72 min, and 7% NaOH (wheat straw); 120 °C, 118 min, and 9% NaOH (corn stalks).	[64]
	Wheat straw	**EM** Cold alkaline extraction and enzymatic hydrolysis. Wheat straw was incubated with NaOH for 90 min at 40 °C. Solid fraction washed with DTW and dried. Commercial enzyme Ultraflo L, Shearzyme 500 L, and Pentopan mono conc. were tested. Straw was sterilized, and 1 g was incubated at 230 rpm for 54 h, with 200 U of endo-xylanase and 42 mL of buffer.	[65]
**Hyaluronic acid** (C**_14_**H**_21_**NO**_11_**)**_n_** **Up to 20,000 kDa**	*Gallus gallus*	**EM** Frozen rooster combs were ground and incubated in acetone for 2 h at 4 °C. Acetone was drained, and dried combs were incubated in 5% CH_3_COON for 2 h and 10 times. EtOH was added, and the solution purified in chloroform 4 times and then chloroform–amyl alcohol. The solution was dialyzed and precipitated with EtOH. **PFN** Chloroform and chloroform–amyl alcohol (1:2) were used in the extraction to remove impurities, such as proteins.	[66]
	Recombinant *Corynebacterium glutamicum*	**EM** Recombinant *C. glutamicum* was cultured in fermenter containing 5 g/L of (NH_4_)_2_SO_4_, 5 g/L of urea, 1 g/L of K_2_HPO_4_, 1 g/L of KH_2_PO_4_, 250 mg/L of MgSO_4_, and 10 mg/L of CaCl_2_. The culture was harvested and purified to isolate the hyaluronic acid. **PFN** First purification with EtOH:C_3_H_8_O (1:1, 1:2, and 1:3) at 4 °C or −20 °C and then centrifuged (4000 rpm, 30 min). Pellets dissolved in 3% SDA, and concentration of HA was calculated. (1) Impurities were removed in three steps consisting of SDA/charcoal/centrifugation; (2) TCA 100% was added to the SDA and incubated in an ice bath for 30 min and centrifuged (16,000 rpm, 30 min); (3) solution was dissolved in chloroform–butanol and stirred on a shaker for 30 min. The aqueous phase was collected.	[67]
**Chitosan** **(C_6_H_11_NO_4_)_n_** **Up to 1000 kDa**	*Penaeus semisulcatus*	**EM** Dry shells demineralized in 1N HCl at 30 °C for 6 h and deproteinized in 3.5% NaOH at 65 °C for 2 h. Then, shells were decolorized in 0.315% NaCl and obtained chitin deacetylated in 50% NaOH at 100 °C for 5 h to obtain chitosan.	[68]
	*Hermetia illucens* larvae, pupal exuviae and dead adults	**EM** Powdered samples were demineralized in 0.5 M formic acid at RT for 1 h, filtered, washed, dried, and then incubated in 2 M NaOH at 80 °C for 2 h for deproteinization. Unbleached and bleached chitin fractions were resuspended in 12 M NaOH and stirred at 100 °C for 4 h, followed by incubation in 1% (*v*/*v*) CH_3_COOH at RT for 48 h. Chitosan was precipitated in pH 8. **PFN** Realized after deproteinization. Bleached chitin was obtained after treating the samples with 5% H_2_O_2_ at 90 °C for 30–60 min. High-purity chitin was obtained (84–86.8%), resulting in higher yields of chitosan.	[69]
**Carrageenan** **(C_24_H_36_O_25_S_2_)_n_** **Up to 2000 kDa**	*Hypnea* *musciformis*	**EM** Conventional and ultrasonic-assisted extraction (UAE). Sun-dried algae was hydrated with water and kept at RT for 12 h, followed by depigmentation using methanol:acetone (1:1). The depigmented extract was treated with 3% KOH at 80 °C for 4 h and filtered/washed. Retained was redissolved in water, incubated at 90 °C for 4 h, and then filtered. Extract was precipitated with 95% ethanol, dried, and milled. UAE was performed as conventional until depigmentation then sonicated at 400 and 500 W for 10 and 20 min in an ice bath.	[70]
	*Eucheuma cottonii*	**EM** Two extraction methods using Ca(OH)_2_ and a second with NaOH were performed. Dry algae were mixed with Ca(OH)_2_ and incubated at 80–110 °C for 2–12 h and filtered; CO_2_ was then injected into extract until pH reached 7–8, twice, and injected again until solution became clear. Finally, 5% KCl was added, and the solution was filtered, dried, and crushed. In the NaOH method, dry algae were incubated in 10% NaOH/12% KCl at 65 °C for 4 h and washed 3 times. Then, algae were incubated in water for 4 h and boiled at 100 °C for 4 h. After that, 5% KCl solution was used to precipitate carrageenan.	[71]
**Ulvan** **RMO (C_5_H_9_O_4_)_n_** **XYL (C_6_H_10_O_7_) _n_** **UA (C_5_H_8_O_6_) _n_** **Up to 200 kDa**	*Cotesia glomerata* and *Ulva flexuosa*	**EM** Dry algae extracted by Soxhlet for 7 h with DCM/acetone (1:1). The residual was subjected to a hot extraction with water in a water bath for 7 h at 75–85 °C. Extract was filtrated and liquid concentrated for enzymatic hydrolysis with α-amylase at 20 °C for 30 min. Then, extract was treated with proteinase K at 37 °C for 24 h, and the reaction stopped by heating at 80 °C for 10 min. Ethanol was added and incubated for 24 h in a fridge to precipitate ulvan. **PFN** Washed consecutively with ethanol, acetone, and diethyl ether.	[72]
	*Ulva armoricana*	**EM** Enzyme-assisted extraction. Crushed algae were incubated with water (1:1) and a neutral endo-protease, a mix of neutral and alkaline endo-proteases, a multiple mix of carbohydrates, a mix of endo-1,4-β-xylanase/endo-1,3(4)-β-glucanase, cellulase, and exo-β1,3(4)-glucanase enzymes (6% weight/dw, w/dw) at 50 °C for 3 h, followed by denaturation at 90 °C for 15 min. Samples were filtered and then freeze-dried.	[73]
**Alginate** **MA (C_6_H_8_O_6_)_n_** **GA (C_6_H_8_O_7_)_n_**	*Sargassum vulgare*	**EM** Dry algae were soaked in 2% formaldehyde for 24 h, washed with water, and left for 24 h in 0.2M HCl. Second wash with water and extraction with 2% Na_2_CO_3_ at 60 °C for 5 h and precipitation of alginate with ethanol. **PFN** Re-precipitation with EtOH and acetone.	[74]
	*Sargassum baccularia*, *Sargassum binderi, Sargassum siliquosum* and *Turbinaria conoides*	**EM** Both a cold and hot extraction were performed. In the cold extraction, algae were left in 1% CaCl_2_ at RT for 18 h, washed with DTW, and incubated with 3% Na_2_CO_3_ for 1 h and left for 18 h. Mixture was separated by centrifugation and alginate extracted by adding EtOH/water (1:1). Precipitate was collected and dried. In the hot extraction, algae were treated similarly as in cold method, except the incubation time and temperature were 3 h and 50 °C. **PFN** Washed with EtOH and dried in air, followed by drying in a vacuum oven.	[75]
**Fucoidan** **(C_6_H_10_O_5_S)_n_** **Up to 1000 kDa**	*Sargassum wightii*	**EM** Hot-water extraction and ultrasonication-assisted extraction.	[76]
	*Ecklonia maxima, Laminaria pallida* and *Splachnidium rugosum*	**EM** 3 methods using a hot-water extraction (24 h at 70 °C), acid extraction (0.15 M HCl at 65 °C for 2 h), and salt extraction (incubated with methanol–chloroform overnight, RT) were performed.	[77]
**Glucomannan** **GLU (C_6_H_10_O_5_)_n_** **MAN (C_6_H_12_O_6_)_n_**	*Dendrobium officinale*	**EM** Solid–liquid extraction. Petroleum ether (70 °C/2 h), 80% EtOH (90 °C/2 h), and DTW (100 °C/2 h). **PFN** Precipitated with EtOH, centrifuged, resuspended in distilled water and deproteinized 5 times (Sevag reagent), washed with ethyl acetate/acetone and EtOH.	[78]
	*Dendrobium* *devonianum*	**EM** Reflux system with methanol for 1 h at 65 °C and microwave-assisted extraction (800 W/80 °C/9 min), precipitation with 95% EtOH, and redissolved in water (60 °C). **PFN** Ultra-centrifugal filtration.	[79]

## Data Availability

Not applicable.

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
