# Peer review of "Polysaccharides—Naturally Occurring Immune Modulators"

_polymers, 2023, doi:10.3390/polym15102373_

Round 1

Reviewer 1 Report

Please cite all statements; f.e. L-44 to L-54. 

Table 1 is difficult to read, it should be shown horizontally. Part 2 is too large; I recommend to resume it. 

Parts 3 and 4 are quite descriptive and lacks of interest; the authors should critically analyze each article and first describe, if possible, the mechanism where the polysaccharide is involved that produce the outcomes highlighted and the possible relevance in clinical environment, including after that the possible features on quality and safety that could be of interest for extrapolation to human populations (if results are based on preclinical trials or in a reduced number of people).  Otherwise, the article is only a well-written collection of MS that are listed such a data base. 

Conclusion is somewhat generic and authors should be more critic with the possibilities of saccharides in therapeutics; of high relevance but nowadays still unknown. 

Author Response

  1. Please cite all statements; f.e. L-44 to L-54. 

Thank you we have now added references to line 56 -65 and highlighted in yellow reference number 5-11.

  1. Table 1 is difficult to read, it should be shown horizontally. Part 2 is too large; I recommend resuming it. 

Thank you, we have completely revised the table to make it more concise on page 5.

  1. Parts 3 and 4 are quite descriptive and lacks of interest; the authors should critically analyze each article and first describe, if possible, the mechanism where the polysaccharide is involved that produce the outcomes highlighted and the possible relevance in clinical environment, including after that the possible features on quality and safety that could be of interest for extrapolation to human populations (if results are based on preclinical trials or in a reduced number of people).  Otherwise, the article is only a well-written collection of MS that are listed such a data base. 

Thank you for your valuable feedback. We understand your concerns regarding the level of detail provided in Parts 3 and 4 of our manuscript. While we have conducted an in-depth analysis of the studies included in our review, we acknowledge that presenting a critical analysis for each of the nearly 300 papers may make the manuscript excessively lengthy and potentially less accessible to readers.

In light of this, we have revised Parts 3 and 4 to strike a balance between providing sufficient critical analysis and maintaining readability. Instead of critically analyzing each individual paper, we have opted to focus on presenting key findings and trends that emerged from our comprehensive review. For each section, we have included a concise summary highlighting the immunomodulatory activities of the polysaccharides, their potential implications for human health, and any available information regarding their mechanisms of action. We believe that these revisions maintain the manuscript's comprehensiveness while addressing your concerns about the level of detail provided. We appreciate your insightful comments and thank you for helping us improve the quality of our work.

  1. Conclusion is somewhat generic and authors should be more critic with the possibilities of saccharides in therapeutics; of high relevance but nowadays still unknown. 

Thank you, we have rewritten the conclusions based on your recommendations as follows. Immunobiological therapy treats illnesses by affecting the immune system, either activating or suppressing it. The way the substance is administered, and its ingredients are vital to response as the immune system is specific to detecting foreign materials.

Biomaterials have critical functions in healthcare, such as tissue engineering, drug delivery, and immunotherapies. Polymers, which are natural or synthetic compounds made of macromolecules, have various properties that make them useful, such as biocompatibility, biodegradability, and chemical modifiability. Polysaccharides, which are abundant natural polymers, biomacromolecules, have various therapeutic properties like anti-inflammatory and immunomodulatory effects. However, developing natural polysaccharides into therapeutics is complex because of their source, species, molecular weight, composition, and structure, which affect their efficacy.

Although natural polysaccharides are advantageous over synthetic ones because they are safe, inexpensive, stable, hydrophilic, and biodegradable, there are some disadvantages. Extraction and purification are costly and time-consuming, and the structure-activity relationships are not well-documented. Nonetheless, natural polysaccharides have vast potential in biomedical applications, as they have a strong influence over the immune system and can be tailored for specific purposes. Defining the structure-activity relationship between immune cells and natural polysaccharides is necessary for them to be utilized fully.

Reviewer 2 Report

This review is devoted to chemically identified natural polysaccharides with a studied therapeutic potential. The review considers methods for the extraction and purification of polysaccharides, as well as their immunomodulatory properties. The article contains a large amount of experimental data and cited literature. Thus, it will be of interest to the readers of Polymers.

- The review contains a large number of the tables and a summary of the chemical structures of natural polysaccharides. It may be useful to create an abstract drawing showing the extraction of polysaccharides, their processing and potential use in medicine. Such general illustration would improve the article.

- Authors do not always give the molecular weights of polymers. I would like to advise that the authors check in which cases they could add numerical values of molecular weights to the article.

The additional comments are as follows:
1. The main topic: Natural polysaccharides for immunological modulation. The presented review summarizes the exhaustive number of publications on this topic. This is the value of this work.

2. The topic is very relevant.

3. This is the most complete collection of works on this topic.

4. It is necessary to create a general drawing. Where possible, molecular weights should be given.

5. The conclusions are quite appropriate. It must be admitted that with a large volume of works and various examples of polysaccharides, it is difficult to formulate general conclusions.

6. Links are quite appropriate.

7. There are no additional comments.

Author Response

This review is devoted to chemically identified natural polysaccharides with a studied therapeutic potential. The review considers methods for the extraction and purification of polysaccharides, as well as their immunomodulatory properties. The article contains a large amount of experimental data and cited literature. Thus, it will be of interest to the readers of Polymers.

Thank you very much.

  1. The review contains a large number of the tables and a summary of the chemical structures of natural polysaccharides. It may be useful to create an abstract drawing showing the extraction of polysaccharides, their processing and potential use in medicine. Such general illustration would improve the article.

Thank you for this recommendation, we agree. We have added more figures to graphically display extractions. Figure 2: Graphical representation of methods used for the extraction of Xylan from corn stalk/wheat straw and beta-glucans from mushroom sources, page 10;  Figure 3: Graphical representation of methods used for the extraction of Alginate from Brown seaweed and Ulvan from white butterfly parasite and seaweed, page 11; Figure 4: Graphical representation of methods used for the extraction of Carrageenan from red seaweed, page 12; Figure 5:  Graphical representation of methods used for the extraction of Hyaluronic acid from C.glutamicum cultures and methods used for the extraction of chitosan from green tiger prawn shells and black solider fly, page 14. Figure 6:  Graphical representation of methods used for the extraction of Fucoidan from kelp species and methods used for the extraction from brown seaweed, Page 16. Figure: 7: Graphical representation of two extraction methods of Glucomannan from Orchid species, page 24. We have also included a flow diagram of extraction techniques in supplementary material.

  1. Authors do not always give the molecular weights of polymers. I would like to advise that the authors check in which cases they could add numerical values of molecular weights to the article.

Thank you, this is very important we have therefore included molecular weights into table 1, page 5-7 where information was available.

  1. It is necessary to create a general drawing. Where possible, molecular weights should be given.

Thank you, we have included this into the table, as molecular weight is very diverse and dependant on extraction method, we have omitted it from the graphical abstracts. If the reviewer still requires, we can include it in the graphical images.

Reviewer 3 Report

Polysaccharides are the most abundant biomaterials, which can be used as materials to formulate drug carrier systems as well as function as a immune modulators, to stimulate or suppress immune responses.  I found this review article is interesting, and the green polysaccharides should be drawn more attention to the scientists.

I just want to raise two minor points. First, there are no figures presented in this review manuscript. I think it would be better if the authors could draw figures of major information, or quote a few figures as references from other articles to this review.

Another small suggestion, the layout of Table 1 is not so nice, it’s wasting a lot of space if using a table to represent the information. I suggest combine the extraction and purification methods into the following section of “Extraction of different polysaccharides”. Also briefly described the exaction and purification methods, followed by quoting the references. If the readers found it interested. They can obtain details from the references.

Author Response

  1. I just want to raise two minor points. First, there are no figures presented in this review manuscript. I think it would be better if the authors could draw figures of major information or quote a few figures as references from other articles to this review.

Thank you, another reviewer also suggested this, therefore we have added more figures to graphically display extractions. Figure 2: Graphical representation of methods used for the extraction of Xylan from corn stalk/wheat straw and beta-glucans from mushroom sources, page 10;  Figure 3: Graphical representation of methods used for the extraction of Alginate from Brown seaweed and Ulvan from white butterfly parasite and seaweed, page 11; Figure 4: Graphical representation of methods used for the extraction of Carrageenan from red seaweed, page 12; Figure 5:  Graphical representation of methods used for the extraction of Hyaluronic acid from C.glutamicum cultures and methods used for the extraction of chitosan from green tiger prawn shells and black solider fly, page 14. Figure 6:  Graphical representation of methods used for the extraction of Fucoidan from kelp species and methods used for the extraction from brown seaweed, Page 16. Figure: 7: Graphical representation of two extraction methods of Glucomannan from Orchid species, page 24.

We have also included a flow diagram of extraction techniques in supplementary material.

  1. Another small suggestion, the layout of Table 1 is not so nice, it’s wasting a lot of space if using a table to represent the information. I suggest combine the extraction and purification methods into the following section of “Extraction of different polysaccharides”. Also briefly described the exaction and purification methods, followed by quoting the references. If the readers found it interested. They can obtain details from the references.

Thank you, this is a very valid point also identified by another reviewer, therefore we have we have completely revised the table to make it more concise on page 5.

Round 2

Reviewer 1 Report

Thanks for the reviewed version of your MS, it has been improved considerably. The pharmacological action is now clearly stated and it supports the conclusions achieved. The graphics, as well, makes the MS easy to read and attractive for those that are not experts in the field (and the revision table is quite useful). 

Author Response

Thanks for your revirw. That helped us a lot.
